



# Ice island thinning: Rates and model calibration with in situ observations from Baffin Bay, Nunavut

Anna J. Crawford[1,2], Derek Mueller[1], Gregory Crocker[2], Laurent Mingo[3], Luc Desjardins[1], Dany Dumont[4], Marcel Babin[5]

[1]Department of Geography and Environmental Studies, Carleton University, Ottawa, Ontario, K1S 5B6, Canada

[2]School of Geography and Sustainable Development, University of St Andrews, St Andrews, KY16 9AJ, United Kingdom

[3]Blue System Integration Ltd., Vancouver, British Columbia, V5W 3H4, Canada

[4]Institut des sciences de la mer de Rimouski, Université du Québec à Rimouski, Rimouski, Québec, G5L 3A1, Canada

[5]Département de Biologie, Université Laval, Québec, Québec, G1V 0A6, Canada

*Correspondence to*: Anna J. Crawford (ajc44@st-andrews.ac.uk)

**Abstract.** A 130 km$^2$ tabular iceberg calved from Petermann Glacier in northwest Greenland on 5 August 2012. Subsequent fracturing generated many individual large "ice islands", including "Petermann Ice Island (PII)-A-1-f", that drifted between Nares Strait and the North Atlantic. Thinning caused by basal and surface ablation increases the likelihood that these ice islands will fracture and disperse further, thereby increasing the risk to marine transport and infrastructure as well as the distribution of freshwater from the polar ice sheets. We use a unique stationary and mobile ice penetrating radar dataset collected over four campaigns to PII-A-1-f to quantify and contextualize ice island surface and basal ablation rates and calibrate a forced convection basal ablation model. The ice island thinned by 4.7 m over 11 months. The majority of thinning (73 %) resulted from basal ablation, but the associated volume loss was ~12 times less than that caused by areal reduction (e.g. wave erosion, calving, and fracture). However, localized thinning may have influenced a large fracture event that occurred along a section of ice that was ~40 m thinner than the remainder of the ice island. The calibration of the basal ablation model, the first with such field data, supports assigning the theoretically-derived value of $1.2 \times 10^{-5}$ m$^{2/5}$ s$^{-1/5}$ °C$^{-1}$ to the model's bulk heat transfer coefficient. Overall, this work highlights the value of systematically collecting ice island field data for analyzing deterioration processes, assessing their connections to ice island morphology, and adequately developing models for operational and research purposes.





# 1 Introduction

Approximately 30 to 60 % of the freshwater flux from the Greenland Ice Sheet is in the form of solid ice discharge, i.e. iceberg calving (Bamber et al., 2012; Enderlin et al., 2014). Particularly large iceberg calving events that occur at the floating
ice tongues in northwest Greenland produce "ice islands" that, through their drift and deterioration, disperse freshwater between Nares Strait and the North Atlantic. These ice islands often ground on the continental shelf of Baffin Island during their journey to southern latitudes (Crawford et al., 2018a). Such groundings can disrupt benthic ecosystems (Dowdeswell and Bamber, 2007), put seafloor infrastructure at risk (Fuglem et al., 2017), extend and stabilize landfast ice covers offshore (Fraser et al. 2012; Massom et al. 2001), and impact the biological and physical composition of ocean waters in their vicinity due to
concentrated meltwater input resulting from their deterioration (Jansen et al., 2007; Stern et al., 2015).

Ice island thinning has previously been estimated via freeboard monitoring with satellite-borne altimetry data (Bouhier et al., 2018; Jansen et al., 2007). Without field observations, the individual contribution of surface versus basal ablation to that thinning can then only be derived through modelling (Ballicater Consulting, 2012; Jansen et al., 2007), often with a relatively simple fluid-dynamics approach being used to model basal ablation in iceberg and ice island studies (Ballicater Consulting,
2012; Merino et al., 2016; Wagner and Eisenman, 2017). This forced convection model is based on the transfer of heat across a flat-plate due to the turbulent flow of the underlying water. A bulk heat transfer coefficient is used to estimate an ablation rate based on the differential velocity between the ice island and the ocean current as well as the water temperature across the length of the ice island (Ballicater Consulting, 2012; Bouhier et al., 2018; Weeks and Campbell, 1973). Previous efforts to calibrate basal ablation models have relied upon remotely sensed datasets as well as potentially inaccurate modelled
environmental data (Bouhier et al., 2018; Jansen et al., 2007). This is largely due to the challenges associated with collecting field data for determining the individual rates of surface and basal ablation.

We overcame these challenges to collect a unique dataset from "Petermann Ice Island (PII)-A-1-f'" between November 2015 and September 2016. PII-A-1-f was grounded on the continental shelf of Baffin Island and had a surface areal extent of 13 km$^2$ when it was first visited. The field work included repeat mobile ice penetrating radar (mIPR) transects and the
deployment of a stationary IPR (sIPR). These were the first such field data to be collected from an ice island or iceberg in either the Arctic or Antarctic. The measurements are used to calibrate the forced convection basal ablation model, which had previously not been validated or calibrated with direct field observations. In addition, the IPR data was used to assess the spatial and temporal variation in ice island thinning and ablation rates. Using remotely sensed imagery to monitor areal reduction, the ablation magnitudes are put into context with respect to other processes (e.g. fracture) that contributed to the
deterioration of PII-A-1-f.

# 2 Study site

PII-A-1-f was a fragment of the 130 km$^2$ PII that calved from the Petermann Glacier in northwest Greenland on 5 August 2012 (Crawford et al., 2018a). After calving, the Canadian Ice Service (CIS; Environment and Climate Change Canada) tracked



the ice island with RADARSAT-2 synthetic aperture radar (SAR) acquisitions. Between August 2012 and November 2014 the
ice island drifted through Nares Strait and Baffin Bay, though it also experienced periods of stagnation while grounded in
Kane Basin and northern Baffin Bay (Fig. 1a). A portion of the deterioration that PII-A-1-f experienced was caused by sidewall
notches that progressively enlarged on opposing sides of the ice island. The notch 'roots', or the tips of these wedge-shaped
features, were located in the vicinity of a linear surface feature that was identified in a ScanSAR acquisition in November 2012
(Fig. 1b). This deterioration was likely caused by increased wave-induced turbulent heat flux within these wedge-shaped
features as described by White et al. (1980).

The linear surface feature and one sidewall notch were still apparent when PII-A-1-f became grounded in November 2014
at 67°23'N, 63°18'W, approximately 10 km from the east coast of Baffin Island and 35 km southeast from the Hamlet of
Qikiqtarjuaq, Nunavut (Fig. 1a). Ice island groundings are especially common in the region immediately north of the
Cumberland Peninsula due to the presence of many underwater shoals and ridges. PII-A-1-f was visited at this location by four
field teams with transportation provided by the CCGS *Amundsen* icebreaker and its helicopter during the annual ArcticNet
science cruises in October 2015, July 2016 and September 2016. An additional field campaign was completed in May 2016
when a field team accessed the ice island, then surrounded by sea ice, by snowmobile from Qikiqtarjuaq, Nunavut.

## 3 Methods

### 3.1 Thinning (temporal assessment)

To ascertain the magnitudes and rates of surface and basal ablation, a series of ablation stakes, a 40 MHz sIPR, and a small
meteorological station with a sonic ranger and camera were installed on PII-A-1-f on 20 October 2015 (Fig. 2). The ice island
position was recorded hourly with a Garmin 6X-HVS GPS (Garmin International, Inc.) and air temperature ($T_a$) was measured
with a 109 thermistor (Campbell Scientific Canada Corporation (CSCC)) in a radiation shield at 1-minute intervals and logged
as hourly averages. All meteorological station data were recorded on a CR1000 datalogger (CSCC) and telemetered with an
Iridium L-Band modem (9522B; CSCC).

Ice thickness was recorded at a resolution of 0.67 m and telemetered daily by the sIPR (Blue System Integration, Ltd.) until
27 September 2016. Full details regarding this system set-up and measurement specifications can be found in S1 and Mingo
et al. (forthcoming). The ice thickness data were linearly interpolated between dates when measured changes in thickness were
recorded (16 November 2015 - 18 September 2016). These dates were used to establish "calibration intervals" for the
calibration of the forced convection basal ablation described in the following section.

Daily mean surface ablation was calculated from hourly SR50A sonic ranger (CSCC) height-above-surface (snow or ice)
measurements. Surface ablation was also calculated from five ablation stakes that were marked with tape every 10 cm and
placed between the meteorological station and the sIPR. A weekly image acquired with a Campbell Scientific CC5MPX
camera that was controlled by the meteorological station to visually check on the positioning of the sIPR system also captured
these ablation stakes. No snow was present on the date when the stakes and instruments were installed and the average weekly





ablation (or accumulation) at these stakes was calculated with ImageJ (v. 10.8.0) software using the markings for scale (Abramoff et al., 2004). Increases in surface height due to the accumulation of snow are presented in this study as negative values. A linear interpolation was applied to estimate daily ablation or accumulation between these observations so that the sIPR, SR50 and ablation stake data were available over a consistent time span. Basal ablation was calculated by subtracting

the interpolated ablation magnitudes derived from the stake data from the ice thickness time series. These values are therefore approximations of daily basal ablation rates, as they were based on values of ice thickness that were linearly interpolated across longer time periods. Finally, the ablation stake data were used instead of the SR50 data, since surface conditions were visually confirmed to be similar to those near the sIPR (10-15 m away), whereas this was not monitored for the SR50 (30 m from the sIPR).

**3.2 Basal ablation model calibration**

**3.2.1 Oceanographic data collection and comparison**

The derived basal ablation was used to calibrate the forced convection model described in the Introduction. In addition to the general paucity of ice island thinning measurements, there is a large dearth of *in situ* oceanographic data available for the validation and calibration of iceberg and ice island numerical deterioration models. Here, we use two oceanographic field

datasets to validate and calibrate a time-series of modelled ocean temperature, salinity and current data that spans the full duration for which basal ablation rates were available. Using the oceanographic model data greatly extended the time span over which the model calibration could be conducted.

The first oceanographic field dataset was collected by the CCGS *Amundsen* during its annual ArcticNet science research cruise. Conductivity, temperature, depth (CTD) profiles were conducted around PII-A-1-f on 28 July (4 casts) and 29

September (5 casts) 2016 (Amundsen Science Data Collection, 2016) (Fig. 1c,d). CTD profiles were also collected on 29 July 2016 (2 casts) and 28 September 2016 (1 cast) close to the location of a sea ice camp that was situated 20 km northwest (67° 29' N, 63° 47' W) of the grounding location of PII-A-1-f (Fig. 1a bottom inset). This sea ice camp was operated by the GreenEdge project which was based out of Université Laval, Québec (http://www.greenedgeproject.info/). The second oceanographic field dataset was composed of 39 CTD profiles (SBE 49 FastCAT CT; Sea-Bird Electronics, Inc.) acquired at

the GreenEdge sea ice camp between 20 April and 22 July 2016. Absolute salinity ($S_A$; g kg$^{-1}$) and conservative temperature ($\Theta$; °C) measurements were reported at 1 m depth bins for all CTD casts. Current speed ($u$; m s$^{-1}$) was calculated as an average of measurements of the current vectors taken every 30 sec over 2 m depth bins and recorded every 30 minutes at the ice camp with a Teledyne/RDI Workhorse Sentinel 300 kHz acoustic Doppler current profiler (ADCP; Teledyne RD Instruments). A linear interpolation was applied to fill missing values before daily average $u$ was calculated for the individual depth bins.

Data from the Copernicus Marine Environment Monitoring Service (CMEMS) Global Ocean Physical Reanalysis (GLORYS12V1) product were used for the calibration of the forced convection basal ablation model. This reanalysis product, referred to as the "CMEMS data" henceforth, had a spatial resolution of 1/12° and 50 depth levels. Potential temperature,





practical salinity and the horizontal current velocity components were extracted from the model grid cells that corresponded with the locations of PII-A-1-f and the GreenEdge sea ice camp at the model depth bin in which the keel of the ice island was

located over the course of sIPR data collection. Keel depth was calculated assuming ice density ($\rho_i$) = 873 kg m$^{-3}$ (Crawford et al., 2018c) and hydrostatic equilibrium. Potential temperature and practical salinity were converted to $\Theta$ and $S_A$, respectively, using the Gibbs Sea Water functions provided in the R 'gsw' package (Kelly, 2017).

The CMEMS data was compared against the *in situ* oceanographic data to justify its use for calibrating the forced convection basal ablation model and to identify any bias in the modelled oceanographic data. Comparisons were conducted between: 1)

the full CCGS *Amundsen* CTD profiles collected at the location of PII-A-1-f and near the GreenEdge sea ice camp location in July and September 2016, 2) the full CCGS *Amundsen* profiles and the CMEMS data profiles, and 3) the CMEMS data time series and the mean $S_A$, $\Theta$ and $u$ values of all CTD casts and ADCP measurements that fell within the CMEMS depth bin in which the ice island keel was located. Bias in the CMEMS data was identified as consistent over- or under-estimation of the GreenEdge sea ice camp time series of $S_A$, $\Theta$ and $u$. If bias existed in a given variable, the CMEMS data was corrected by the

average daily difference between it and the *in situ* values.

### 3.2.2 Model calibration

The forced convection basal ablation model (Eq. (1)) estimates an ablation rate ($M_b$; m d$^{1}$) from the velocity difference between the iceberg and the water ($\Delta u$; m s$^{-1}$) and the driving temperature ($\Delta T$) across the length ($L$; m) of an ice face using a bulk heat transfer coefficient, $C$ (m$^{2/5}$ s$^{-1/5}$ °C$^{1}$) (Weeks and Campbell, 1973). In the calibration, it was assumed that the ice

island was stationary and therefore $\Delta u$ is simply equivalent to $u$. $\Delta T$ was calculated as the difference between the ocean temperature at the keel depth and the melting point of ice (equivalent to the freezing point of the adjacent sea water). The melting point ($M_p$; °C;) was adjusted to account for the influence of meltwater near the ice-water interface with an established empirical relationship with the far field water temperature ($\Theta$) and the freezing temperature of the far field ocean water ($\Theta_f$; °C; Eq. (2)) (Kubat et al., 2007; Løset, 1993). $\Theta_f$ was derived with $S_A$ and pressure ($p$; dbar) as per TEOS-10 conventions (IOC,

SCOR and IAPSO, 2010).

$$M_b = C\Delta u^{\frac{4}{5}}\frac{\Delta T}{L^{\frac{1}{5}}} \qquad (1)$$

$$M_p = \Theta_f e^{-0.19(\Theta - \Theta_f)} \qquad (2)$$

Values of $C_i$ were obtained for each calibration interval $i$, which are associated with a measured change in ice thickness. The individual $C_i$ values were calculated by dividing the cumulative basal ablation over the respective calibration interval by the corresponding cumulative driving force (i.e., $\sum \Delta u^{\frac{4}{5}}\frac{\Delta T}{L^{\frac{1}{5}}}$). The driving force was derived from CMEMS modelled oceanographic data ($S_A$, $\Theta$, and $u$) at the ice island keel. $p$ was set to the pressure at the keel depth. $L$ was assigned as the



median of all distances between the sIPR location and all vertices of the outline of the areal surface extent of PII-A-1-f that
was digitized from RADARSAT-2 Fine-Quad (FQ; 8 m nominal resolution) SAR imagery acquired on 27 July 2016 (S2).

A second calibration for each interval was conducted with the CMEMS data after corrections were applied based on the
comparisons against the *in situ* oceanographic data (Section 3.2.1). A final calibration of $C$ was obtained based on the basal
ablation and driving force over the *total* duration over which basal ablation was derived. This final calibration provided a
single value of $C$ as opposed to the previous calibrations where values of $C_i$ were obtained for each calibration interval.

An analysis was conducted to determine the sensitivity of the basal melt magnitude predicted by the forced convection basal
ablation model to variations in $u$, $\Theta$, and $C$. The two variables and one parameter were individually perturbed across 8 equally
spaced intervals that covered their observed and calculated ranges. The non-perturbed variables were held constant at their
median values. All assigned values were based on the corrected CMEMS data series. The sensitivity of the forced convection
basal ablation model was assessed as the average percent increase in cumulative basal ablation predicted for the duration over
which basal ablation was derived with each incremental increase in the value assigned to a given variable/parameter. The
sensitivity was analyzed over this longer time period due to the certainty in the bulk basal ablation derived from total ice
thinning and surface ablation.

### 3.3 Thinning (spatial assessment)

A repeat transect was conducted to assess spatial variation in thickness and thinning and to verify that the sIPR data was
representative with respect to other locations across the ice island. An initial thickness profile was collected with a 25 MHz
mIPR (Blue System Integration Ltd.) that was towed by snowmobile over an approximately 3 km transect on 8 May 2016. A
series of 10 ablation stakes were also installed along the profile route. Thickness data were collected over 2.4 km of the original
transect when the same mIPR system was towed on foot on 28 September 2016. Eight of the stakes were also re-measured at
this time. Further details of the mIPR system can be found in S1 and Mingo and Flowers (2010).

All mIPR processing was conducted with Radar Tools (release 0.4), a library of Python scripts and tools that was used to
standardize, clean, visualize, and process data contained in the raw radar data (S1; Wilson, 2013). This program was also used
to select the location of the air and reflected radar waves. Ice thickness was calculated with Eq. (3) (Wilson, 2012):

$$H = \sqrt{\left(\left(t + \frac{s}{v_a}\right)\frac{v}{2}\right)^2 - \frac{s^2}{4}} \tag{3}$$

where $H$ is thickness (m), $s$ is the distance between the transmitting and receiving antennas (m), $t$ is the time between the
recording of the air and reflect waves by the receiver, and $v_a$ is the speed of the electromagnetic wave in air ($3.0 \times 10^8$ m s$^{-1}$)
traveling between the transmitter and receiver (Wilson, 2012). The speed of the radar wave in ice ($v$) was set to $1.7 \times 10^8$ m s$^{-1}$
(Macheret et al., 1993). The ice thickness resolution ($\pm$ 0.5 m) was limited by $v$ and the waveform sampling interval of the
mIPR system (Crawford, 2013).





Snow was present on the ice surface when the mIPR transect was conducted in May 2016. However, an insufficient number
of snow depth measurements were recorded to adequately account for the snow depth over the transect area and it was not
possible to distinguish the snow/ice interface in the radargram. Thickness (ice + snow) in May was therefore calculated using
a single velocity value, $v = 1.7 \times 10^8$ m s$^{-1}$. However, an additional uncertainty was added to the May ice thickness to account
for the possible presence of snow. This uncertainty was based on the mean snow depth (36 cm) recorded at nine locations
along the mIPR transect and the amount of time a radar wave would travel through this layer given $v = 2.0 \times 10^8$ m s$^{-1}$ for snow
(Haas and Druckenmiller, 2009). The errors associated with the resolution of the mIPR system and the average snow depth
estimate were summed to determine the average amount that the ice thickness, as measured in May, could be overestimated
(0.8 m). Uncertainty in thickness change calculations was determined by propagating the uncertainties in ice thickness resulting
from the resolution of the mIPR system and the presence of snow.

The magnitude of thinning that occurred between May and September 2016 was calculated between the closest pairs of
mIPR traces recorded during each field visit. To improve positional accuracy, the GPS locations recorded by the mIPR onboard
GPS were replaced with those recorded by a Hiper V dual-frequency GPS (Topcon Corp.) after precise point positioning (PPP)
processing (Natural Resources Canada, 2016). The September transect was adjusted relative to the orientation of PII-A-1-f in
May by matching mIPR traces known to be collected at the same location (i.e., the ablation stakes) on the ice island to correct
for a small amount of movement by the ice island between the two field visits. Since it was not possible to retrace the transect
exactly due to changes in topography between field visits, the thickness measurements that were compared between May and
September were offset by varying distances (meters to 10s of meters). Using Eq. (4), we derived an index of comparability for
these measurements from the percentage of overlap between the radar footprints at the ice-water interface. The radii ($r$) of
these footprints depend on the center wavelength of the 25 MHz antenna ($\lambda = 6.8$ m), relative permittivity of ice ($K$; 3), and $H$
(Leucci et al., 2003),

$$r = \frac{\lambda}{4} + \frac{H}{\sqrt{K+1}}. \tag{4}$$

The positions of thickness change measurements were categorized as ≥ 50 % overlap, ≥ 30 % overlap, < 30 % overlap, and no
overlap.

### 3.4 Surface extent reduction and contributions to deterioration

The areal extent of PII-A-1-f was monitored with seven RADARSAT-2 FQ SAR scenes to determine the relative importance
of surface and basal ablation to the total deterioration of the ice island. The SAR scenes were acquired between 1 November
2015 and 23 September 2017, and the image processing and areal extent digitization details are included in S2. We quantified
and compared the contributions of basal ablation, surface ablation, and areal reduction processes (e.g. fracture, forced and
buoyant convection, wave erosion, and calving) to the overall deterioration of PII-A-1-f over the temporal extent sIPR data
collection. We also estimated the contribution of surface and basal ablation to total deterioration over the longer temporal
extent that RADARSAT-2 FQ images were acquired. For this estimation, a cubic spline was applied to fill missing cumulative





ablation values between 24 September 2016 and 21 October 2016 and it was assumed that the same surface and basal ablation magnitudes that occurred in the first year of observation also occurred between October 2016 and September 2017.

## 4 Results

### 4.1 Thinning (temporal assessment) and environmental conditions

PII-A-1-f decreased in thickness by $4.7 \pm 1.4$ m over the eleven months that sIPR data was collected and by $4.0 \pm 1.4$ m during the length of time that basal ablation was estimated and used for the basal ablation model calibration (Fig. 3a; Table 1). The latter can be divided into three periods related to surface ablation magnitudes as per the ablation stake data (Fig. 3b; Table 1). These are referred to as "ablation periods" for the remainder of the text. Minimal surface ablation was observed over ablation period 1 (November to mid-December 2015). No surface ablation occurred during the second period, which
distinguishes it from ablation periods 1 and 3. Therefore, basal ablation was the sole contributor to thinning during ablation period 2 when the mean daily thinning rate was 0.9 cm d$^{-1}$ (mid-December 2015 to mid-July 2016). This rate tripled during the third ablation period, which spanned from mid-July to September 2016, due to the onset of continuous surface ablation in mid-July. Ninety-eight percent of the recorded surface ablation occurred in this period when $T_A$ was consistently $> 0$ ℃. $T_A$ was $> 0$ ℃ for 15 % of the second ablation period, however, this contributed to melting the snow that had accumulated on the
ice surface instead of ice ablation (Fig. 3b; Table 1).

The mean daily basal melt rate ($M_b$) between November 2015 and September 2016 was almost three times greater than the mean rate of daily surface ablation ($M_s$; cm d$^{-1}$). However, this ratio varied between ablation periods. For example, the mean $M_b$ was seven times greater than the mean $M_s$ during ablation period 1 when only 2 cm of surface ablation was observed. In contrast, the mean $M_s$ was approximately 50 % greater than the mean $M_b$ over the third ablation period when over 1 m of
surface ablation was observed over the two-month duration (Fig. 3b; Table 1).

Basal ablation was responsible for 73 % of the observed thinning between November 2015 and September 2016. The mean $M_b$ increased between successive ablation periods (Table 1) and the increase in daily mean $M_b$ from 0.9 to 1.2 cm d$^{-1}$ between periods 2 and 3 coincides with an increase in CMEMS temperature and velocity.

### 4.2 Basal ablation model calibration

#### 4.2.1 Oceanographic data comparisons

Figure 4 shows the CMEMS $\Theta$, $S_A$ and $u$ for the duration that PII-A-1-f basal ablation was derived. The mean $\Theta$ and $S_A$ of measurements from CTD casts acquired in the vicinity of PII-A-1-f or the GreenEdge sea ice camp are plotted alongside the model data in Fig. 4a,b. These data are associated with the depth interval of the model data in which the keel of the ice island fell within for the duration of the data collection. The mean $u$ in this depth interval, as measured by the ADCP moored near to
the GreenEdge sea ice camp location, is included in Fig. 4c. The data collected at the GreenEdge sea ice camp ($\Theta$ and $S_A$) and





at the nearby moored ADCP ($u$) provide the longest *in situ* time series of oceanographic conditions and were used to assess if these variables were consistently over- or under-estimated in the CMEMS data.

The means of the absolute daily difference between $\Theta$ and $S_A$ at the GreenEdge sea ice camp and the CMEMS data were 0.07 °C and 0.67 g kg⁻¹, respectively. The CMEMS $S_A$ was consistently greater than the *in situ* data and was therefore corrected

by subtracting 0.67 g kg⁻¹ (Fig. 4b, dotted lines). A bias in $\Theta$ was not seen and no correction was applied; however, there was a consistent under-estimation of $u$ in the CMEMS data (Fig 4c). The CMEMS data was corrected by the mean daily difference (0.1 m s⁻¹) between the CMEMS and *in situ* values of $u$ (Fig. 4c, dotted lines).

CTD profiles collected in the vicinity of PII-A-1-f and the GreenEdge sea ice camp on successive days by the CCGS *Amundsen* in July and September 2016 were plotted (not shown) to compare the oceanographic conditions at the two sites. In

the depth interval of interest, the values of $\Theta$ and $S_A$ were reasonably similar and not consistently different between sites. This supported our use of the $S_A$ correction for the CMEMS data that was derived from the difference between the CMEMS data and the longer time series collected at the GreenEdge sea ice camp. This $S_A$ correction was applied to the CMEMS data associated with the location of PII-A-1-f, which was used for the model calibration (Fig. 4b).

The full CMEMS $\Theta$ and $S_A$ profiles were also plotted (not shown) against those acquired by the CCGS *Amundsen* to ensure

that the CMEMS data were generally representative of the water column. The CMEMS profiles were reasonable in form and, in the depth interval of interest, CMEMS consistently over-estimated the $S_A$ recorded by the CTD casts acquired by the CCGS *Amundsen*. The average over-estimation was 0.60 g kg⁻¹. This further supports the recommendation to apply the 0.67 g kg⁻¹ correction to the CMEMS $S_A$ for the forced convection basal ablation model calibration. The CMEMS $\Theta$ was both over- and under-estimated in the depth interval of interest, which reinforces the recommendation to not apply a correction to this variable.

**4.2.2 Model calibration and sensitivity analysis**

$C_i$ values were calculated with the uncorrected and corrected CMEMS data for each of the six calibration intervals. The range and mean values of $C_i$ calculated with the uncorrected CMEMS data were an order-of-magnitude larger than those that have been previously calibrated or theoretically derived (Weeks and Campbell, 1986; White et al., 1980). Due to this, plus the poor representation of uncorrected CMEMS $S_A$ and $u$, it was decided to further analyze the more reasonable $C_i$ values that were

calculated with the corrected CMEMS data. These latter values of $C_i$ are included in Table 2 with the corresponding mean $\Delta T$ and $u$ values found with the CMEMS data.

Using this data, the mean $C_i$ value was $1.7 \times 10^{-5}$ m²ᐟ⁵ s⁻¹ᐟ⁵ °C⁻¹ and values for the individual calibration intervals ranged from 7.8 x 10⁻⁶ to $2.1 \times 10^{-5}$ m²ᐟ⁵ s⁻¹ᐟ⁵ °C⁻¹. The greatest values of $C_i$ were associated with calibration intervals 3 through 6. While intervals 3, 4 and 5 were characterized by relatively low $\Theta$ along with high $S_A$ and $u$ values (Fig. 4), it is difficult to draw

conclusions regarding the alignment of oceanographic conditions and $C_i$ values due to the low resolution of the sIPR thickness measurements. Cumulative basal ablation was over-predicted by 38% when the forced convection basal ablation model was run with the mean $C_i$ value over the 308 days that basal ablation was calculated. Finally, when calculated with the total basal ablation and total driving force between 15 November 2015 and 18 September 2016, $C$ was calibrated to $1.2 \times 10^{-5}$ m²ᐟ⁵ s⁻¹ᐟ⁵ °C⁻¹.





Values of $u$, $\Theta$, and $L$ were not normally distributed. Therefore, the median values of these variables were first assigned
during the sensitivity analysis when a given variable was held constant. The calibrated $C_i$ values were found to follow a normal
distribution, though the power of this test is low due to the small sample size. For consistency, the median $C_i$ value was also
assigned when this parameter was held constant during the sensitivity analysis.

Of the environmental driving variables included in the forced convection basal ablation model, $\Delta T$ has the greatest relative
range of values over all of the calibration intervals (0.10 to 0.37 °C). A mean increase of 19 % was applied to the $\Delta T$ value
during the sensitivity analysis and the same increase in cumulative basal ablation was predicted over the time period that basal
ablation was derived with observations. This is due to the linear relationship between $\Delta T$ and $M_b$ in Eq. (1) (Table 3).

A linear relationship also exists between $C$ and $M_b$ in Eq. (1); however, the range in $C_i$ was more constrained relative to the
$\Delta T$ data series. For this reason, the cumulative predicted basal ablation only increased by 16.5 % with each incremental change
in the value assigned to $C$ during the sensitivity analysis (Table 3). Finally, due to the smaller range of $u$ (0.11 to 0.18 m s$^{-1}$)
and the non-linear relationship between $u$ and $M_b$ in Eq. (1), the relative increment adjustment (6.1 %) and corresponding
change (4.8 %) to the predicted cumulative basal ablation over the time period that basal ablation was derived with observations
is less than that associated with $\Delta T$ or $C$.

### 4.3 Thinning (spatial assessment)

The data collected at the main site on PII-A-1-f over 11 months provide unprecedented information regarding the temporal
thinning of an ice island. However, these data are representative of a single point on a large ice island. Considering that the
rate of surface ablation along the mIPR transect would have been near zero until the snow had melted around 15 July 2016, as
determined from the ablation stakes at the sIPR site, the surface would have ablated at a rate of 1.5 cm d$^{-1}$ after this date. This
is slightly less than the rate at the sIPR site over ablation period 3 (1.6 cm d$^{-1}$; Table 1) and increases the confidence that the
surface ablation conditions at the sIPR site were similar to those across the mIPR transect.

The ice island thickness over the mIPR transect ranged from 80.7 to 127.1 [-0.5,+0.8] m in May and 77.3 to 123.7 ± 0.5 m
in September 2016 (Fig. 5a). A 50 m long section of relatively thin ice that was approximately 80 m thick was recorded along
transect segment AB during both field visits (Fig. 5a); this section was approximately 40 m thinner than adjacent areas. The
air and basal wave locations in the mIPR data could not be definitively selected when this thin section was passed during
transect segment CD (Fig. 5a). This made it impossible to ascertain thickness at this location; however, a pair of in-line
hyperbolae in both the May and September 2016 radargrams support the interpretation that this thinner section ran across both
transect segments AB and CD. This section of thin ice is referred to as a 'subsurface feature' and corresponds to the location
of the linear 'surface feature' described in Sect. 2. Together these are referred to as the 'paired feature'.

The magnitude of thinning (3 to 4 m) observed over this thin section observed along transect segment AB (shown in the
grey boxes in Fig. 5) was in the low to mid-range of the thinning observed along the entire transect. However, the general
gradient in thinning magnitudes leading towards the paired feature shows a similar pattern in both transect segments AB and
CD (shown in the pink boxes in Fig. 5). Thinning magnitudes of 3 to 4 m increased to 5 to 6 m across a distance of



approximately 90 to 150 m, ending at the surface feature and where a gap in the thickness record exists. A fracture that occurred along this paired feature in September 2017 caused the areal extent of the ice island to reduce by approximately $2.7 \pm 0.1$ km$^2$.

## 4.4 Volume/area loss and contributions to deterioration

The volume of PII-A-1-f was $1.4 \pm 0.01$ km$^3$ when it was first visited in October 2015. By September 2016, the volume and areal extent decreased by $0.4 \pm 0.01$ km$^3$ and $3.4 \pm 0.1$ km$^2$, respectively. A fracture event in September 2016 caused approximately 94 % of the areal reduction and 88 % of this volume loss. These values represent the maximum possible reductions caused by the fracture. Other areal reduction processes (e.g., small-scale calving) would also have contributed to areal reduction during the time interval between RADARSAT-2 acquisitions over which the fracture occurred.

The 23 month RADARSAT-2 monitoring period (October 2015 to September 2017) captured a longer period of areal change. During the latter half of this time span (i.e., September 2016 and September 2017), the ice island twice un-grounded and re-grounded in the same vicinity and the large September 2017 fracture event occurred (Fig. 1e). The ice island volume decreased by $0.6 \pm 0.01$ km$^3$ over this period, with the vast majority of this volume loss (94 %) being caused by processes that decreased the areal extent of the ice island. The remaining volume was lost to basal and surface ablation, with basal ablation

causing three times more volume loss than surface ablation. These ablation magnitudes were extrapolated from the on-ice data collection period. Over the entire two years that PII-A-f was monitored via RADARSAT-2 image acquisition, areal reduction processes reduced the volume of the ice island by approximately 67 %. Combined, surface and basal ablation resulted in an approximate 7 % reduction the ice island volume, with basal ablation again causing three times more loss than surface ablation.

## 5 Discussion

**5.1 Basal ablation model calibrations**

    Two approaches are predominately used to model ice island and iceberg basal ablation (Bouhier et al., 2018). The first is the semi-empirical fluid-dynamics approach that is calibrated in this study; Eq. (1) approximates melt resulting from the bulk energy transfer occurring within the complex boundary conditions that are present at the ice-water interface (Weeks et al., 1973; White et al., 1980). This model has been widely utilized for iceberg and ice island deterioration modelling in both the

Arctic (Bigg et al., 1997; Ballicater Consulting, 2012; Keghouche et al., 2010; Wagner and Eisenman, 2017) and Antarctic (Gladstone et al., 2001; Martin and Adcroft, 2010; Merino et al., 2016). The second approach is based on thermodynamic principles (Bouhier et al., 2018). Holland and Jenkins (1999) and Hellmer and Obers (1989) document a more complex three-equation thermodynamic model for ice shelf basal ablation, which represents both the salt and temperature flux across the ice-ocean interface.

The forced convection basal ablation model is advantageous due to its computational simplicity and the direct incorporation of $\Delta u$. In the thermodynamic approach, the values assigned to the turbulent heat and salt exchange parameters must be adjusted for varying values of $\Delta u$ in iceberg and ice island applications (Jansen et al. 2007). Jansen et al. (2007) provide calibrated



values for three stages of drift, each with a unique range of $\Delta u$ values for an Antarctic ice island. While Eq. (1) in the forced convection model directly incorporates $\Delta u$, model skill would likely improve if Eq. (2) was calibrated for drifting vs. grounded

ice islands. Equation (2) was developed to account for the plume of relatively cold and fresh iceberg meltwater that has been observed to surround icebergs and inhibit further melt (Foldvik et al., 1980). This meltwater plume will be stripped from the keel as $\Delta u$ increases, as is the case when an ice island is grounded (Jansen et al., 2007). Therefore, different parameterizations of Eq. (2) are likely required for predicting the basal ablation of drifting versus grounded ice islands. It is possible that the adjustment to the melting point of ice ($M_p$) to account for the influence of the meltwater plume is not necessary and $M_p$ will

simply equal the far field ocean temperature ($\Theta_f$). Determining this will require concerted study of the difference in the basal boundary layer conditions of grounded versus drifting ice islands. Observations of $\Delta u$ for the drifting ice island case are rare but would be useful for this work and for correctly assigning values to this variable in Eq. (1).

This study used a unique dataset of observed ice island thinning, as well as *in situ* and modelled oceanographic conditions to present a calibration of the forced convection basal ablation model. It is noted that the exponents within Eq. (1) could also

be calibrated. However, these have remained constant in previous iceberg and ice island literature while the bulk heat transfer coefficient, $C$, has typically been assigned one of two values that differ by an order of magnitude. Bigg et al. (1997), Bouhier et al. (2018), Martin and Adcroft (2010), Wagner and Eisenman (2017), and Weeks and Campbell (1973) assign a value of $6.74 \times 10^{-6}$ $m^{2/5}$ $s^{-1/5}$ $°C^{-1}$. However, confidence interval of the normal distribution of our calibrated $C$ values does not overlap with this theoretically-derived value of the bulk heat transfer coefficient. We recommend assigning the $C$ parameter a value of

$1.2 \times 10^{-5}$ $m^{2/5}$ $s^{-1/5}$ $°C^{-1}$ when modelling ice island basal ablation in the future. This value was calculated from the calibration conducted over the full time span that basal ablation data was available and not the individual calibration intervals associated with the measured changes in ice thickness. In addition, the confidence interval of the normal distribution of $C_i$ values found in Table 2 does overlap with this value. The value also matches, remarkably, that which was theoretically derived by White et al. (1980) and was assigned in the ice island deterioration model used by the CIS (Ballicater Consulting, 2012). We believe

that it is more appropriate to use this value instead of the mean of the individually calibrated $C_i$ values reported in Table 2 (1.7 x $10^{-5}$ $m^{2/5}$ $s^{-1/5}$ $°C^{-1}$), as we cannot be sure of the alignment between changes in oceanographic conditions, basal melt rates, and $C_i$ values due to the low resolution of data collected by the sIPR. It is noted that Crawford et al. (2018b) assigned a value of 1.3 x $10^{-5}$ $m^{2/5}$ $s^{-1/5}$ $°C^{-1}$ to $C$, supplied by Crawford (2018), when quantifying the distribution of freshwater input from ice island melt through the eastern Canadian Arctic. The use of this previous value would cause a very minimal skew in the distribution

of freshwater input, slightly overestimating the freshwater input at the higher latitudes in their study region.

The *in situ* dataset of oceanographic conditions in the vicinity of PII-A-1-f and the GreenEdge sea ice camp was of paramount importance for validating and correcting the CMEMS data for bias. However, the calibration could be further improved by obtaining a longer time series of oceanographic data in closer proximity to the ice island. In general, there is a paucity of *in situ* oceanographic data collected in the vicinity of ice islands. Collection of such data will allow for further,

improve drift and deterioration analyses of icebergs and ice islands. Modifying the sIPR to resolve smaller magnitudes of



thickness change is also recommended. This would make it possible to relate these higher-quality thinning measurements to corresponding surface ablation and recorded oceanographic conditions.

**5.2 Ablation rates and contributions to overall deterioration**

Field measurements of ice island thinning are extremely sparse. Prior to this study, the spot values reported by Halliday et al. (2012) from a 17 km$^2$ drifting in the Labrador Sea were the only known field observations of ice island thinning. The PII-A-1-f thinning dataset greatly improves on this previous work, as the long-term sIPR time-series and repeat mIPR transects together produce a comprehensive dataset that allows us to assess the spatial and temporal variations in thinning and ablation. It would be highly beneficial to re-deploy the sIPR on a drifting ice island in the future to 1) augment the number of observations of ice island thinning and 2) begin comparing basal ablation occurring to drifting vs. grounded ice islands.

The average $M_b$ of 3.4 cm d$^{-1}$ reported by Halliday et al. (2012) was greater than that observed for PII-A-1-f. Since the basal ablation rate for a drifting ice island should be lower than that of a grounded ice island due to decreased $\Delta u$ and the protection of a meltwater plume, this greater melt rate was likely due to the higher $\Theta$ off the coast of Labrador. Elevated ocean temperatures also contributed to the high, 13.5 m month$^{-1}$ basal ablation rate that was estimated for the grounded Antarctic ice island, "A38-B". Basal ablation was reported to cause 96 % of the thinning of this ice island that had a surface extent of approximately 7600 km$^2$ (Jansen et al., 2007), whereas 73 % of the thinning of PII-A-1-f was a result of this process.

While basal ablation was responsible for the majority of the thinning of PII-A-1-f, this process caused the ice island volume to reduce by approximately 5 %. However, basal ablation indirectly influences further deterioration by reducing the relative thickness of the ice island and decreasing fracture resistance (Goodman et al., 1980; Jansen et al., 2005). This will happen more quickly if an ice island is grounded, as basal ablation could increase by a factor of three after an ice island stops freely drifting (Jansen et al., 2007). Future research, including finite element modelling (e.g., Sazidy et al. submitted), is warranted to assess if and how thinning contributed to the September 2017 calving event. This fracture occurred along the paired feature that was substantially thinner than adjacent ice surfaces and which emanated from the root of the sidewall notch. We also recommend that further research be conducted into the relationship between the presence of the paired feature, the propagation of the sidewall notches before and during the time that PII-A-1-f was grounded in southern Baffin Bay, and the ultimate September 2017 fracture. The enlargement of sidewall notches, similarly to cusped deterioration patterns observed by Ballicater (2012), is a recurring deterioration mechanism that is not considered in deterioration studies or models at this point in time.

**6 Conclusion**

This study focuses on the thinning of PII-A-1-f, an ice island that originated from a calving event at the Petermann Glacier in 2012 and was grounded in western Baffin Bay over a 2-year monitoring time span. A unique field dataset was collected





over four visits between October 2015 and September 2016 and was used to report ice island thinning and ablation rates and calibrate the popular forced convection basal ablation model.

The time series of ice island thinning recorded with a customized sIPR showed the ice island thinned by 4.7 m over the 11 months that on-ice data. Basal ablation was responsible for 73 % of the observed thinning and caused the volume of PII-A-1-f
to decrease by an estimated 5 % over the duration that the ice island was remotely monitored via RADARSAT-2. This was a relatively small change in comparison to the 67 % volume reduction that resulted from processes that contributed to reducing the areal extent of the ice island. However, PII-A-1-f was likely more susceptible to fracture than if it had been freely drifting, due to enhanced basal ablation and a correspondingly faster reduction in relative thickness.

It is important to model ice island basal ablation accurately for predicting the impact of meltwater input on the ocean system
(Jansen et al., 2007). Additionally, basal ablation will alter the relative thickness of an ice island, which will influence fracture likelihood (Goodman et al., 1980), drift patterns (Barker et al., 2004) and grounding locations (Sackinger et al., 1991). This is the first study to calibrate the forced convection basal ablation model for ice island or iceberg use with field data of ice island thinning, which removed uncertainty regarding estimated ablation rates from remotely sensed datasets. The calibrated value of the bulk heat transfer coefficient ($1.2 \times 10^{-5}$ m$^{2/5}$ s$^{-1/5}$ °C$^{-1}$) is in-line with the larger of two values assigned in previous iceberg
and ice island basal ablation models, and we recommend that this value be used in future modelling endeavours. It is important to conduct such field studies to develop and validate methods for modelling ice island thickness change (i.e., surface and basal ablation), as this will inform future deterioration investigations and improve ice island drift and deterioration forecasting in both of the polar regions (Barker et al., 2004). The calibration of the forced convection basal ablation model might also be used to generally predict when grounded ice islands might thin enough to drift free, assuming certain shoal bathymetry and ice
island morphology. This may be especially useful along the eastern coast of Canada where shipping and offshore industry operates, and where ice island grounding is a common occurrence. Overall, the work presented in this study highlights the value of systematic ice island field data collection. This is necessary for deterioration analyses, connecting morphology and deterioration, and developing high-quality models for operational and research purposes.

*Author contribution* AJC, GC and DM were responsible for project design. LM designed and produced the stationary and
mobile ice penetrating radars and assisted in data processing and interpretation. AJC carried out field work and data analyses. DD and MB provided oceanographic data associated with the GreenEdge sea ice camp. DM, GC, LD, LM and DD contributed to the preparation of the manuscript, which was led by AJC.

*Competing interests* The authors declare that they have no conflict of interest.

*Data availability* Amundsen CTD data is available from the Polar Data Catalogue (PDC) under the file identifier 12713. Data
collected at PII-A-1-f by the authors (i.e., data associated with ablation stakes, the weather station and sonic ranger, sIPR and



mIPR) is also available from PDC (CCIN 13091), as is the RADARSAT-2 imagery (geotiffs) and digitized polygons of the areal extent of PII-A-1-f. CMEMS products are available at http://marine.copernicus.eu/. Data associated with the GreenEdge project are available, either through download or correspondence with a project coordinator, at http://www.greenedgeproject.info/.

*Acknowledgements* We would like to acknowledge the field assistance of Jaypootie Moesesie, Graham Clark, Jill Rajewicz, Jonathan Gagnon, Luke Copland, Abigail Dalton, Lauren Candlish, Hugo Jacques, Tom Desmeules, Natalie Theriault, Alison Cook, Alexis Burt, Leah Braithwaite, Julie Payette, Claire Bernard-Grand'Maison, Eric Brossier and Christian Haas. Adam Garbo and Iain Burnett helped with field preparation. This unique dataset would not be possible to collect without the access provided through the collaboration of ArcticNet and the CCGS *Amundsen*. We thank Canadian Coast Guard (CCG) captains

Alain Lacerte and Alain Gariépy as well as Transport Canada pilots Olivier Talbot, Alain Roy and Guillaume Carpentier for providing safe transport to our field site. We also greatly thank the entire CCG crew, who were instrumental in our field success. The hard work of ArcticNet staff, including Keith Lévesque, Phillipe Archambault, Louis Fortier, Jean-Éric Tremblay, Tim Papakyriakou, Leah Braithwaite, Annisa Merzouk and Colleen Gombault deserves great recognition. We thank Doug King, Geography and Environmental Studies, Carleton University and Douglas MacAyeal, University of Chicago for reviewing an

earlier version of the manuscript.

Instrument development and field work were funded by the Northern Transportation Adaptation Initiative of Transport Canada, the Polar Knowledge Canada Safe Passage project (#1516 – 065) and Polar Knowledge Canada's Northern Scientific Training Program. A.J. Crawford received personal funding from the Garfield Weston Foundation, the Natural Sciences and Engineering Research Council (Canada) and Environment and Climate Change Canada. This study has been conducted using

E.U. Copernicus Marine Service Information. RADARSAT-2 scenes were acquired through a joint partnership agreement between Water and Ice Research Lab and the Canadian Ice Service, Environment and Climate Change Canada. RADARSAT data and products are © MacDonald, Dettwiler and Associates Ltd. (2010-2017), all rights reserved. RADARSAT is an official mark of the Canadian Space Agency. Some of the data presented herein were collected by the Canadian research icebreaker CCGS *Amundsen* and made available by the Amundsen Science program, which was supported by the Canada Foundation for

Innovation and Natural Sciences and Engineering Research Council of Canada. The views expressed in this publication do not necessarily represent the views of Amundsen Science or that of its partners. Some of the data presented in this analysis were collected by the GreenEdge project. The GreenEdge project is funded by the following French and Canadian programs and agencies: ANR (Contract #111112), CNES (project #131425), IPEV (project #1164), CSA, Fondation Total, ArcticNet, LEFE and the French Arctic Initiative (GreenEdge project). The GreenEdge project would not have been possible without the support

of the Hamlet of Qikiqtarjuaq and the members of the community as well as the Inuksuit School and its Principal Jacqueline Arsenault. The GreenEdge project is conducted under the scientific coordination of the Canada Excellence Research Chair on Remote sensing of Canada's new Arctic frontier and the CNRS & Université Laval Takuvik Joint International laboratory (UMI3376). The GreenEdge field campaign was successful thanks to the contribution of J. Ferland, G. Bécu, C. Marec, J.



Lagunas, F. Bruyant, J. Larivière, E. Rehm, S. Lambert-Girard, C. Aubry, C. Lalande, A. LeBaron, C. Marty, J. Sansoulet, D.

Christiansen-Stowe, A. Wells, M. Benoît-Gagné, E. Devred and M.-H. Forget from the Takuvik laboratory, C.J. Mundy and

V. Galindo from University of Manitoba as well as F. Pinczon du Sel and E. Brossier from Vagabond. The GreenEdge project

also thanks Michel Gosselin, Québec-Océan, the CCGS *Amundsen* and the Polar Continental Shelf Program for their in-kind

contribution in polar logistic and scientific equipment.



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





**Table 1: Ablation and total thinning magnitudes and rates per ablation period.** Air temperature ($T_a$) data for each period is also included. Inconsistencies between the summed ablation and thinning magnitudes or between the ratios of surface : basal ablation magnitudes versus rates are due to rounding. $M_s$ and $M_b$ represent daily surface and basal ablation rates, respectively.

| Ablation period | Dates | Surface ablation (cm) & $M_s$ (cm d$^{-1}$) | Basal ablation (cm) & $M_b$ (cm d$^{-1}$) | Total thinning (cm) & *rate* (cm d$^{-1}$) | Mean $T_a$ (°C) | Days $T_a >$ 0°C |
|---|---|---|---|---|---|---|
| 1 | 2015-11-16: 2016-12-04 | 2 *(0.1)* | 14 *(0.7)* | 16 *(0.8)* | -13 | 3 |
| 2 | 2016-12-05: 2016-07-14 | 0 | 200 *(0.9)* | 200 *(0.9)* | -15 | 33 |
| 3 | 2016-07-15: 2016-09-14 | 106 *(1.6)* | 78 *(1.2)* | 184 *(2.8)* | 3 | 65 |
| 1-3 | 2015-11-16: 2016-09-14 | 108 *(0.4)* | 295 *(1.0)* | 403 *(1.3)* | -11 | 101 |







**Table 2: Calibrated values for $C_i$ for the individual calibration intervals associated with each measured change in ice thickness.** The corresponding mean driving temperature ($\Delta T$) and velocity ($u$) values, found with the CMEMS data, are also provided.

| Calibration Interval | Date range | $C_i$ (m$^{2/5}$ s$^{-1/5}$ °C$^{-1}$) | $\Delta T$ (°C) | $u$ (m s$^{-1}$) |
|---|---|---|---|---|
| 1 | 2015-11-15 : 2016-01-30 | 7.8 x 10$^{-6}$ | 0.32 | 0.12 |
| 2 | 2016-01-30 : 2016-05-07 | 8.3 x 10$^{-6}$ | 0.24 | 0.12 |
| 3 | 2016-05-07 : 2016-07-09 | 2.6 x 10$^{-5}$ | 0.12 | 0.11 |
| 4 | 2016-07-09 : 2016-08-01 | 2.1 x 10$^{-5}$ | 0.18 | 0.11 |
| 5 | 2016-08-01 : 2016-08-22 | 2.1 x 10$^{-5}$ | 0.17 | 0.11 |
| 6 | 2016-08-22 : 2016-09-18 | 1.6 x 10$^{-5}$ | 0.23 | 0.14 |




**Table 3: Descriptive statistics and sensitivity analysis results.** The key driving variables in the forced convection basal ablation model were individually perturbed in 8 equal increments based on the variable's range. The sensitivity of the model was assessed by the mean percent increase in predicted cumulative basal ablation over the time period that basal ablation was derived with each incremental increase in the value assigned to a given variable.

| | Range | Increment | Mean increase in variable (%) | Mean increase in cumulative basal ablation (m) | Mean increase in cumulative basal ablation output (%) |
|---|---|---|---|---|---|
| $\Delta u$ (m s$^{-1}$) | 0.11 : 0.18 | 0.01 | 6.1 | 0.22 | 4.8 |
| $\Delta T$ (°C) | 0.10 : 0.37 | 0.03 | 18.6 | 0.59 | 18.6 |
| $C$ (m$^{2/5}$ s$^{-1/5}$ °C$^{-1}$) | $7.8 \times 10^{-6}$ : $2.6 \times 10^{-5}$ | $2.3 \times 10^{-5}$ | 16.5 | 0.56 | 16.5 |


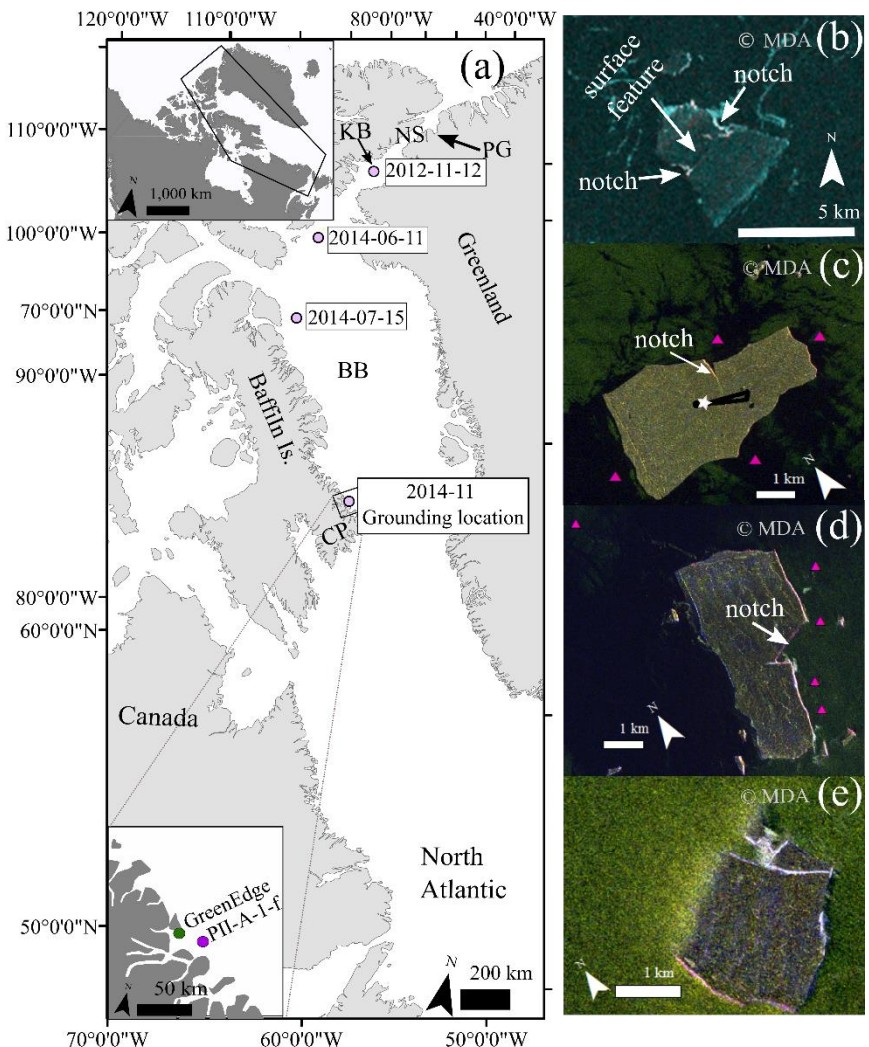

**Figure 1: Remote monitoring of PII-A-1-f and data collection locations.** (a) Dates and locations of "Petermann Ice Island (PII)-A-1-f" as observed with RADARSAT-2 synthetic aperture radar (SAR) observations while the ice island drifted from the Petermann Glacier (PG), through Nares Strait (NS), Kane Basin (KB) and Baffin Bay (BB) before grounding north of the Cumberland Peninsula (CP). The polygon in the top inset shows the location of the larger-scale map. The bottom inset shows the proximity of the grounding location to the GreenEdge sea ice camp. (b) RADARSAT-2 ScanSAR image (100 m nominal resolution) showing the surface feature and sidewall notches. (c) – (e) RADARSAT-2 Fine-Quad SAR (8 m nominal resolution) scenes acquired on 27 June 2016, 27 September 2016 and 23 September 2017, respectively. Conductivity, temperature, depth (CTD) cast locations are denoted by triangles in (c) and (d) and the star in (c) denotes the location of the stationary ice penetrating radar and weather station; the ~3 km mobile IPR transect conducted in May 2016 is shown as a black line. All RADARSAT-2 images are presented as colour composites (ScanSAR polarizations: red = HH, blue and green = HV. Fine-Quad polarizations: red = HH, green = VV, blue = HV).



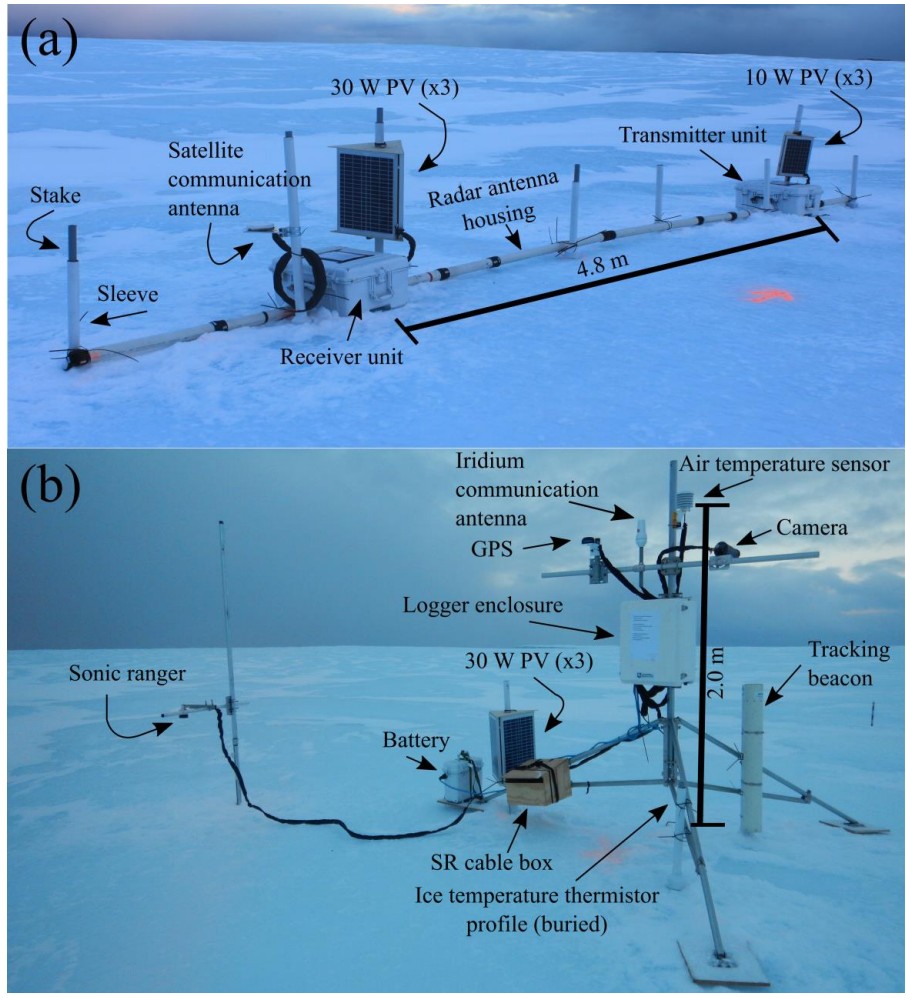

**Figure 2: Instruments installed on Petermann Ice Island-A-1-f on 20 October 2015.** (a) Stationary ice penetrating radar (sIPR) components deployed with antennas 'in-line' and attached to eight sleeves that slid over stake anchors. (b) Automatic weather station components, including a camera that acquired a weekly image of the sIPR and ablation stakes (not shown). The two systems were installed 30 m apart on two small ridges at the location denoted in Fig. 1c. PV = photovoltaic panel, SR = sonic ranger.

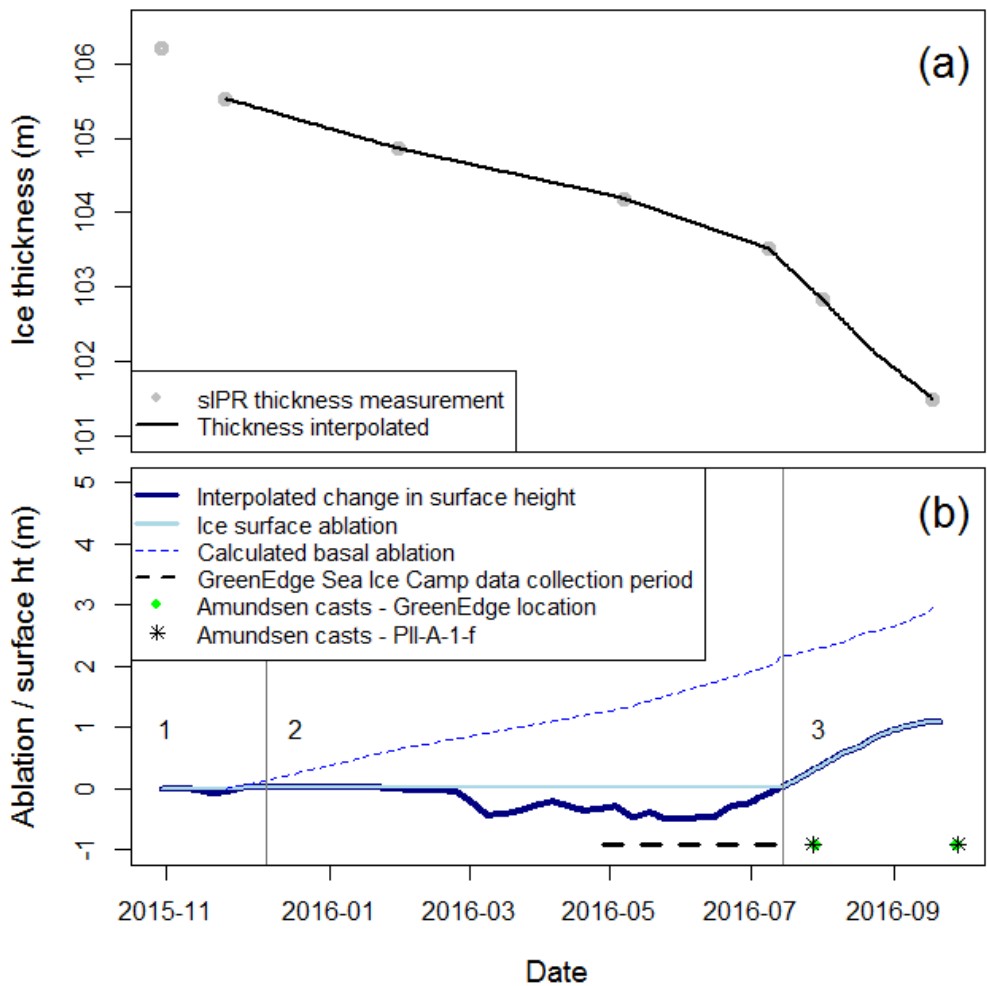


**Figure 3: Ablation and thickness change.** (a) "Petermann ice island (PII)-A-1-f" thickness change as measured by the stationary ice penetrating radar (sIPR) (grey dots) and interpolated between thickness measurements (black line). The first thickness observation was not included in the interpolation due to uncertainty imposed by sIPR measurement settings. (b) Surface and basal ablation, as well as the change in surface height (ht) relative to the start of sIPR data collection. The latter, represented by the thick blue line, includes ice ablation (positive

values) as well as snow accumulation (negative values). Basal ablation is calculated with the linearly interpolated thickness values in a) and should be taken as an estimate of daily magnitudes. The vertical lines denote three surface ablation periods that correspond with the periods described in Table 1. The timing of CTD profiling by the CCGS *Amundsen* and the time span associated with oceanographic data collection at the GreenEdge sea ice camp are also denoted.


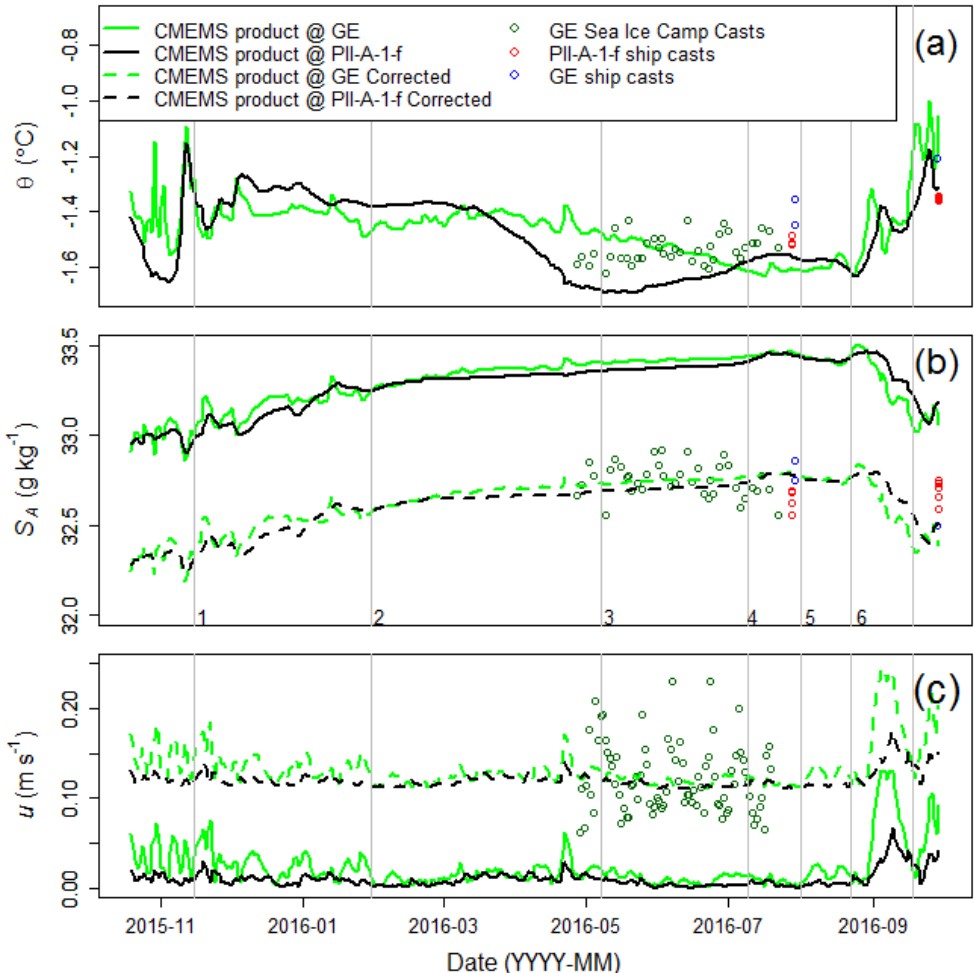


**Figure 4: Oceanographic data collected or modelled over the time span that PII-A-1-f basal ablation rates were derived.** CMEMS product data for the grid cells in which "Petermann Ice island (PII)-A-1-f" (solid black lines) and the GE sea ice camp (solid green lines) were located: (a) conservative temperature ($\Theta$), (b) absolute salinity ($S_A$), and (c) velocity ($u$). As PII-A-1-f was grounded, $u$ is equivalent to the differential velocity between the ice island and the ocean current. Field data collected by the CCGS *Amundsen* and the GreenEdge project are shown in the respective panels. The CMEMS data, corrected for bias in $S_A$ and $u$, are shown as dotted lines in (b) and (c), respectively. The vertical lines denote the calibration intervals, numbered in (b), associated with thickness change measurements that were used for the calibration of the forced convection basal melt model.

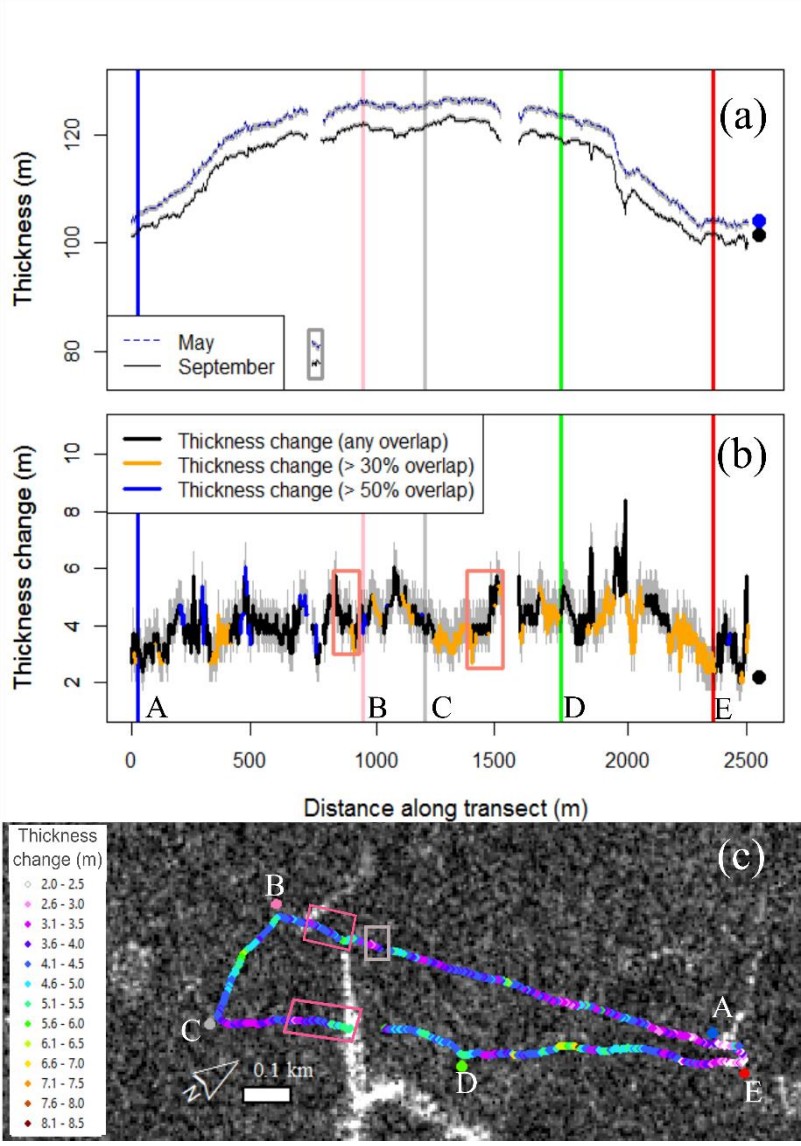

**Figure 5: Spatial variation in morphology and thinning between May and September 2016.** Upper-case letters are used to denote transect segments referred to in the text. (a) Thickness observations collected with mobile ice penetrating radar. The grey box corresponds with the thin ice section also denoted in (c). b) Thickness change. The pink boxes indicate thickness change gradients also identified in (c). Thinning data were categorized based on the amount of overlap between the May and September radar footprints. Grey shading in (a) and (b) denotes uncertainty in thickness measurements and thinning amounts, respectively. The larger, single points denote the respective thickness and thinning measured with corresponding data collected by the sIPR at the main site. (c) Thickness change displayed on a Fine-Quad RADARSAT-2 synthetic aperture radar scene acquired on 28 September 2016 (shown in greyscale). A surface feature is apparent as a line of high backscatter (white) pixels starting at the sidewall notch. The geolocation error of the RADARSAT-2 scene was evaluated and minimal error in the position of the scene was observed.