# Peer review of "Ice island thinning: Rates and model calibration with in situ observations from Baffin Bay, Nunavut"

_The Cryosphere, 2019_

## Author Comment (AC1) · 9 Jul 2019

Greg Crocker is incorrectly affiliated with the University of St Andrews in the original submission of this manuscript. Please note that Greg Crocker is affiliated with the Department of Geography and Environmental Studies, Carleton University, Ottawa, Ontario, K1S 5B6, Canada.

---

## Referee Comment (RC1) · Anonymous Referee #1 · 27 Sep 2019

**General comments**

The article "Ice island thinning: Rates and model calibration with in situ observations from Baffin Bay, Nunavut" presents thickness and ablation rates data collected from an ice island, and uses those measurements to calibrate a coefficient present in an equation used to calculate iceberg basal melt in numerical models. The paper is in general very well-written and brings some much-needed iceberg observations to the scientific community. My biggest concern is the definition of temperature gradient used here, which differs from the traditional $T_{ocean} - T_{ice}$. If ice temperature was taken into

consideration, their calibrated value for C would probably be an order of magnitude smaller. Because of that, I recommend this paper for publication in The Cryosphere after major revisions.

**Specific comments**

1. Lines 28-29: *"Approximately 30 to 60 % of the freshwater flux from the Greenland Ice Sheet is in the form of solid ice discharge, i.e. iceberg calving"*

    Actually, solid ice discharge is not equivalent to iceberg calving. As described by Bamber et al. (2018, JGR Oceans 123(3),1827-1837), "Solid ice discharge is the product of surface velocity and ice thickness along an outlet glacier flux gate, typically located near and upstream of the grounding line". While possibly most of it will be calved, part of this solid ice will be subjected to submarine melting as well.

2. Section 3.1:

    (a) You mention first that ice thickness was recorded daily. Then you say the ice thickness was linearly interpolated between dates used as calibration intervals (6 of them). Why do you have only 6 points for daily data spanning almost a year? Does that mean that the thickness recorded daily had step-changes from one day to another, say, measuring 105.5 m constantly from December 2015 until it suddenly changed to about 105 m in February 2016 (Fig 3a)? I imagine this is due to the instrument precision (0.67 m),

but it would be good to clarify this.

(b) Did you compare the sonic ranger measurements of surface ablation to the stakes' data? I imagine retrieving ablation rates from the stakes has a degree of uncertainty that maybe the SR data could constrain.

3. Section 3.2.2:

   (a) If $M_b$ is in m/day and the other variables have units in seconds, you need to include, on the right-hand side of equation 1, a multiplying factor of 86400.

   (b) You calculate $\Delta T$ as the difference between the ocean temperature at the keel depth (ocean temperature) and the **melting point** of ice. Why? The original equation defines $\Delta T$ as the difference between the ocean temperature and ice temperature. While FitzMaurice et al. (2017) assume $T_{ice} = -15°C$, in models, ice temperature is generally assumed -4°C. If you use $T_{ice} = -4°C$ and considering the temperatures you normally have at the keel (as per Fig 4a), you will have $\Delta T = T_{ocean} - T_{ice} = -1.5 + 4 = 2.5°C$ , which is an order of magnitude larger than your usual $\Delta T$ (Table 2). If $\Delta T$, in turn, is one order of magnitude larger, using the same $M_b$, you will get a C one order of magnitude smaller ($10^{-6}$) which is consistent with the value given by Weeks and Campbell (1986).

4. Lines 273-276: *"While intervals 3, 4 and 5 were characterized by relatively low $\Theta$ along with high $S_A$ and $u$ values (Fig. 4), it is difficult to draw conclusions regarding the alignment of oceanographic conditions and $C_i$ values due to the*

*low resolution of the sIPR thickness measurements."*

In fact, the theoretical way to calculate $C$ is through

$$C = \frac{kPr^{1/3}\alpha}{\rho_i \Gamma \nu^m} \tag{1}$$

where $\alpha$ and m are dependent on iceberg shape, k is the thermal conductivity, Pr is the Prandtl number (function of k), $\rho_i$ is the ice density, $\Gamma$ is the latent het of ice and $\nu$ is the kinematic viscosity (see Supporting Information for FitzMaurice et al. 2017). So, although some of these variables are dependent on temperature and salinity, I would not expect a straightforward relationship between $C$ and oceanographic parameters.

5. Lines 315-316: *"The volume of PII-A-1-f was 1.4 $\pm$ 0.01 km$^3$ when it was first visited in October 2015. By September 2016, the volume and areal extent decreased by 0.4 $\pm$ 0.01 km$^3$ and 3.4 $\pm$ 0.1 km$^2$, respectively."* - What was its area by the time of the first visit?

6. Line 345: Take a look at FitzMaurice et al. (2017, Geophys. Res. Lett., 44, 5637-5644, doi:10.1002/2017GL073585). They discuss a new parameterization of iceberg melting due to the influence of attached/detached plumes. This paper is very relevant for your discussion.

7. Lines 348-350: *"It is possible that the adjustment to the melting point of ice ($M_p$) to account for the influence of the meltwater plume is not necessary and $M_p$ will simply equal the far field ocean temperature ($\Theta_f$)"* - If $M_p$ was the far field

temperature and your $\Delta T$ is $T_{keel} - M_p$, then the basal melt would be close to zero! Once again, it makes more sense to use the temperature of ice. Maybe what you had in mind is to use $M_p$ instead of $T_{keel}$, and do $M_p - T_{ice}$ to calculate $\Delta T$.

8. Lines 350-352: *"Determining this will require concerted study of the difference in the basal boundary layer conditions of grounded versus drifting ice islands. Observations of $\Delta u$ for the drifting ice island case are rare but would be useful for this work and for correctly assigning values to this variable in Eq. (1)."* - It would also be useful to have an estimate of the plume's vertical velocity, according to FitzMaurice et al. (2017).

9. Lines 354-355: It is worth mentioning that those exponents are related to the shape of the iceberg (in this case, taken as a flat plate).

10. Lines 386-387: *"and the protection of a meltwater plume"* - Again, it depends on the vertical velocity of the plume in relation to $\Delta u$ (FitzMaurice et al., 2017). You could have a detached plume in a drifting iceberg.

**Technical corrections**

Line 16: *"(...) thereby increasing the risk to marine transport and infrastructure as well as* **[affecting, impacting]** *the distribution of freshwater from the polar ice sheets."*

Lines 19-20: *"The majority of thinning (73 %) resulted from basal ablation, but the associated volume loss was  12 times less than that caused by areal reduction"* - It is

not clear to me if the "associated volume loss" refers to the total thinning or only to the thinning caused by basal ablation.

Lines 34-35: *"(...) and impact the biological and physical characteristics of ocean waters in their vicinity* **due to meltwater input and latent heat uptake** *resulting from their deterioration"*

Lines 57-61: *"PII-A-1-f was a fragment of the 130 km2 PII that calved from the Petermann Glacier in northwest Greenland on 5 August 2012 (Crawford et al., 2018a). After calving, the Canadian Ice Service (CIS; Environment and Climate Change Canada) tracked the ice island with RADARSAT-2 synthetic aperture radar (SAR) acquisitions. Between August 2012 and November 2014 the ice island drifted through Nares Strait and Baffin Bay, though it also experienced 60 periods of stagnation while grounded in Kane Basin and northern Baffin Bay (Fig. 1a)"* - I assume that the one that "calved from Petermann Glacier in northwest Greenland on 5 August 2012" is the PII. In this case, which ice island are you referring to in the next sentences? PII or its fragment PII-A-1-f?

Line 137: *"ablation rate ($M_b$; m d$^{-1}$)"* - there is a minus sign missing from d exponent

Line 139: *"C ($m^{2/5}$ $s^{-1/5}$ $°C^{-1}$)"* - there is a minus sign missing from °C exponent

Line 142: *"melting point ($M_p$; °C)"* - remove semicolon after °C

Equation 1: using fractional exponents makes the layout of this equation confusing (it looks like $\Delta T$ is part of the 4/5). I suggest using the typical notation:

$$M_b = 86400 C \Delta u^{0.8} \frac{\Delta T}{L^{0.2}} \qquad (2)$$

Line 150: I would rephrase that as *"Values of C were obtained for each calibration interval i ($C_i$)"*, since up to that point you have only defined C.

Line 180: I think you meant *"t is the time between the recording of the air and reflected waves by the receiver"*

Line 184: *"However, an insufficient number (...)"* - Since you use "However," again on line 187, you could remove this one without losing any meaning.

Line 195: *"the locations recorded by the mIPR onboard GPS were replaced with those recorded by a Hiper V dual-frequency GPS"* - Remove first "GPS"

Lines 267-268: *"were an order-of-magnitude larger than those that have been previously calibrated or theoretically derived (Weeks and Campbell, 1986; White et al., 1980)."* - I think you should mention here what the theoretically derived value is, so the reader can readily compare it with the following $C_i$ values

Line 308: *"The magnitude of thinning (3 to 4 m) **observed** over this thin section **observed** along transect segment AB"* - Perhaps replace the second "observed" with "present"

Lines 344 and 347: *"(...) model skill would likely improve if Eq. (2) was calibrated for drifting vs. grounded ice islands. (...) different parameterizations of Eq. (2) are likely*

[Figure]

*required for predicting the basal ablation of drifting versus grounded ice islands"* - I would change the first sentence (even remove it) to avoid repeating information a few lines below.

Line 375: improv**ed**

Line 388: *"(...) also contributed to the high, 13.5 m month$^{-1}$ basal ablation rate that* **Jansen et al. (2007)** *estimated (...)"*

Line 409: *"that on-ice data."* - ... were collected (?)

Table 1: Check the "Dates" column on the first and second ablation periods. I believe it should be 2015-12-04 and 2015-12-05 instead of 2016.

Table 3 caption: *"over the time period that basal ablation was derived with each incremental increase in the value assigned to a given variable."* - I found this sentence very confusing

Figure 5 caption, line 642: There is a "(" missing from "b)"

---

## Referee Comment (RC2) · Anonymous Referee #2 · 30 Oct 2019

This paper presents measured ablation rates of a tabular iceberg which calved from the Peterman Glacier, North Greenland, which are then used to calibrate a model for melt rates. The in-situ data set, which comprises repeated ice-penetrating radar surveys, surface mass balance measurements, as well as oceanographic data nearby is very unique, and a valuable contribution to the cryosphere community. Previously, studies on iceberg ablation rates were mostly based on modelling alone, or estimation of ablation rates from satellite altimetry, which is not straight forward because of the unknown surface processes and densification rates of a possible firn layer, which is especially important for the Antarctic icebergs. Thus, I would see it as an excellent contribution to The Cryosphere. I would like to rise a few points which might need

addressing before publication:

1. At some point (section 3.2.2) it is stated that for the calibration of the model it is assumed that delta u is set equal to the ocean current velocity. As the iceberg is grounded during much of the time this might be a valid assumption. Nevertheless, as the iceberg can be easily tracked by remote sensing data, it would have been possible to look at real drift velocities, and how they compare to the ocean currents. Some studies have suggested (e.g. Lichey & Hellmer, 2001, Jounal of Glaciology) that not only the ocean current is responsible for iceberg drift velocity and direction, but also the wind conditions. A comparison / discussion of these parameters would be an improvement.

2. In the introduction it is stated that this study Is the first of its kind, for Arctic and Antarctic icebergs. However, to my knowledge there was a similar study set up for an Antarctic iceberg (Scambos at al., 2008, Journal of Glaciology) which might deserve a mentioning here.

3. In the discussion the big difference between the ration of basal and surface ablation rates from results of a former study is mentioned. When comparing these results it has to be considered the in case of the other study the Antarctic tabular iceberg started off with a firn column, while the Peterman iceberg did not have any snow cover, so in fact a blue-ice surface. If there is a firn column, surface melt water can percolate into the firn and refreeze, so the mass is not immediately lost. While on blue ice it is more likely to run off. The problem of refreezing melt water and firn densification is the biggest contribution to uncertainty for previous studies estimating ablation of tabular icebergs from altimeter data. For this setting it would be immensely helpful to have an in-situ data set like the one presented here. This might be added to the discussion.

---

## Author Comment (AC2) · 15 Dec 2019

**Reply to Reviewer 1 and other author changes: Manuscript #2019-125 by Crawford, Mueller, Crocker, Mingo, Desjardins, Dumont and Babin:** "*Ice island thinning: Rates and model calibration with in situ observations from Baffin Bay, Nunavut*"

Original reviewer comments are in black.

Author replies are in blue.

**General comments**
The article "Ice island thinning: Rates and model calibration with in situ observations from Baffin Bay, Nunavut" presents thickness and ablation rates data collected from an ice island, and uses those measurements to calibrate a coefficient present in an equation used to calculate iceberg basal melt in numerical models. The paper is in general very well-written and brings some much-needed iceberg observations to the scientific community. My biggest concern is the definition of temperature gradient used here, which differs from the traditional $T_{\mathrm{ocean}}$ - $T_{\mathrm{ice}}$. If ice temperature was taken into consideration, their calibrated value for $C$ would probably be an order of magnitude smaller. Because of that, I recommend this paper for publication in The Cryosphere after major revisions.

We greatly appreciate the reviewer's thorough and informed comments regarding our manuscript, as well as the acknowledgement that our work contributes valuable in situ observations that are largely lacking in this field of research. A complete response regarding the $T_{\mathrm{ice}}$ temperature assignment can be found below in our replies to the reviewer's specific comments. Our current approach for determining $T_{\mathrm{ice}}$ is, in fact, appropriate as we carefully demonstrate. However, we provide additional text regarding the approach taken and state the calibration result is specific to this approach. Our paper has been strengthened by addressing this comment, as doing so provided us with the opportunity to thoroughly research regarding how $T_{\mathrm{ice}}$ is assigned in this model. We highlight and important divergence in approach among papers, discuss the implications and make recommendations.

**Specific comments**
1. Lines 28-29: "Approximately 30 to 60 % of the freshwater flux from the Greenland Ice Sheet is in the form of solid ice discharge, i.e. iceberg calving". Actually, solid ice discharge is not equivalent to iceberg calving. As described by Bamber et al. (2018, JGR Oceans 123(3),1827-1837), "Solid ice discharge is the product of surface velocity and ice thickness along an outlet glacier flux gate, typically located near and upstream of the grounding line". While possibly most of it will be calved, part of this solid ice will be subjected to submarine melting as well.

The reviewer is correct, and we thank them for bringing up this important distinction. We have removed the sentence and now begin the Introduction with new subject matter regarding an anticipated future calving event from Petermann Glacier.

2. Section 3.1:
(a) You mention first that ice thickness was recorded daily. Then you say the ice thickness was linearly interpolated between dates used as calibration intervals (6 of them). Why do you have only 6 points for daily data spanning almost a year? Does that mean that the thickness recorded daily had step-changes from one day to another, say, measuring 105.5 m constantly from December 2015 until it suddenly changed to about 105 m in February 2016 (Fig 3a)? I imagine this is due to the instrument precision (0.67 m), but it would be good to clarify this.

Yes, we interpolated between dates when step changes in the ice thickness measurement were recorded. These step changes were due to the set-up of the stationary ice penetrating radar. We have slightly rearranged the information in the paragraph and add the term 'step changes' to clarify this.

(b) Did you compare the sonic ranger measurements of surface ablation to the stakes' data? I imagine retrieving ablation rates from the stakes has a degree of uncertainty that maybe the SR data could constrain.

We have previously compared these measurements. The very similar trend in surface height and derived basal ablation associated with the sonic ranger data provides confidence in the stake measurements and basal ablation calculations. However, we do not feel that we are able to constrain uncertainty in the stake data with the sonic ranger data due to unknown differences in the surface conditions between their locations. Using the weekly photographs, we are confident that the surface conditions between the stationary ice penetrating radar and the stakes match well whereas the surface conditions were not monitored at the location of the SR50. The difference in surface conditions between the sIPR and the SR50 likely caused the unrealistic basal ablation time series derived from the SR50 data. The latter point is now added in the last paragraph of section 3.1.

3. Section 3.2.2:
(a) If Mb is in m/day and the other variables have units in seconds, you need to include, on the right-hand side of equation 1, a multiplying factor of 86400.

The multiplying factor has been added.

(b) You calculate $\Delta T$ as the difference between the ocean temperature at the keel depth (ocean temperature) and the melting point of ice. Why? The original equation defines $\Delta T$ as the difference between the ocean temperature and ice temperature. While FitzMaurice et al. (2017) assume $T_{ice} = -15°C$, in models, ice temperature is generally assumed -4°C. If you use $T_{ice} = -4°C$ and considering the temperatures you normally have at the keel (as per Fig 4a), you will have $\Delta T = T_{ocean} - T_{ice} = -1.5 + 4 = 2.5°C$, which is an order of magnitude larger than your usual $\Delta T$ (Table 2). If $\Delta T$, in turn, is one order of magnitude larger, using the same $M_b$, you will get a $C$ one order of magnitude smaller ($10^{-6}$) which is consistent with the value given by Weeks and Campbell (1986).

Our approach to assigning $\Delta T$ follows that of Løset (1993a) and the iceberg and ice island deterioration modelling of Kubat et al. (2007) and Ballicater Consulting (2012). We acknowledge that another line of literature has assigned -4 °C to $T_{ice}$, and we reviewed this literature in depth after the topic of ice temperature assignment was brought forward. The foundational literature of Weeks and Campbell (1973) and White (1980) do not explicitly state how $T_{ice}$ was assigned in their work, but it can be inferred through their $\Delta T$ and variable descriptions that $T_{ice}$ was assigned a value representative of the ice interface temperature. The assignment of -4 °C to $T_{ice}$ references back to Løset (1993b), where the top-most temperatures (~50 cm depth) on several icebergs approached -4 °C (see Figure 4 in that paper) and 2D model output shows that this temperature is reached within a few meters of the ice-ocean interface (see figures 8 and 9 of that paper). Løset (1993b) does not explicitly mention this temperature or recommendation of its use in the paper itself. Overall, it is a challenge to determine the single correct value to assign to $T_{ice}$, as this should consider the evolving temperature gradient between the ocean interface and iceberg core. We use the method of Løset (1993a), as it is empirically based on work conducted by Josberger (1977) and is also supported by FitzMaurice and Stern (2018). We have added explicitly statement regarding the calibrated coefficient fitting the value assigned to, or approach used to derive, $T_{ice}$ in the Abstract and Conclusion. We also added a paragraph to section 5.1: Basal Ablation Model Calibration that discusses the ice temperature assignment in our work as well as in previous literature. Examples of how our calibrated value of $C$ would differ due to the use of different $T_{ice}$ values are also provided in this location. We appreciate the push to investigate this topic, as it opened an interesting line of discussion regarding best practices for assigning $T_{ice}$.

4. Lines 273-276: "While intervals 3, 4 and 5 were characterized by relatively **low** $\Theta$ along with high $S_A$ and $u$ values (Fig. 4), it is difficult to draw conclusions regarding the alignment of oceanographic conditions and $C_i$ values due to the low resolution of the sIPR thickness measurements."

In fact, the theoretical way to calculate $C$ is through

$$C = \frac{kPr^{\frac{1}{3}}\alpha}{\rho_i \Gamma v^m} \tag{1}$$

where $\alpha$ and $m$ are dependent on iceberg shape, k is the thermal conductivity, Pr is the Prandtl number (function of k), $\rho_i$ is the ice density, $\Gamma$ is the latent heat of ice and $v$ is the kinematic viscosity (see Supporting Information for FitzMaurice et al. 2017). So, although some of these variables are dependent on temperature and salinity, I would not expect a straightforward relationship between $C$ and oceanographic parameters.

We appreciate the point and explanation that is provided. The sentence has been removed.

5. Lines 315-316: "The volume of PII-A-1-f was $1.4 \pm 0.01$ km$^3$ when it was first visited in October 2015. By September 2016, the volume and areal extent decreased by $0.4 \pm 0.01$ km$^3$ and $3.4 \pm 0.1$ km$^2$, respectively." - What was its area by the time of the first visit?

We have added the surface area of the ice island when it was first visited in October 2015.

6. Line 345: Take a look at FitzMaurice et al. (2017, Geophys. Res. Lett., 44, 5637-5644, doi:10.1002/2017GL073585). They discuss a new parameterization of iceberg melting due to the influence of attached/detached plumes. This paper is very relevant for your discussion.

We thank the reviewer for bringing the FitzMaurice et al. (2017) paper to our attention. It is indeed relevant to incorporate into our discussion of iceberg/ice island melt model parameterizations. We have included the following text in the relevant paragraph, "(…) FitzMaurice et al. (2017) showed to be the case when parameterizing Eq. (1) for the sidewall melt of an iceberg with different scenarios of meltwater plume and relative ocean velocities."

7. Lines 348-350: "It is possible that the adjustment to the melting point of ice ($M_p$) to account for the influence of the meltwater plume is not necessary and $M_p$ will simply equal the far field ocean temperature ($\Theta_f$)" - If $M_p$ was the far field temperature and your $\Delta T$ is $T_{keel}$ - $M_p$, then the basal melt would be close to zero! Once again, it makes more sense to use the temperature of ice. Maybe what you had in mind is to use $M_p$ instead of $T_{keel}$, and do $M_p$ - $T_{ice}$ to calculate $\Delta T$.

The reviewer's comment led us to discover an important omission in this sentence. We have updated the sentence to read, "It is possible that the adjustment to the melting point of ice ($M_p$) to account for the influence of the meltwater plume is not necessary and $M_p$ will simply equal the far field ocean **freezing** temperature ($\Theta_f$)". Please see our response to point 3b regarding the ice temperature assignment.

8. Lines 350-352: "Determining this will require concerted study of the difference in the basal boundary layer conditions of grounded versus drifting ice islands. Observations of $\Delta u$ for the drifting ice island case are rare but would be useful for this work and for correctly assigning values to this variable in Eq. (1)." - It would also be useful to have an estimate of the plume's vertical velocity, according to FitzMaurice et al. (2017).

We agree that an estimate of the plume's vertical velocity would be useful for understanding sidewall boundary layer conditions and modeling sidewall melt. However, our work focuses solely on basal boundary conditions and we do not feel that mentioning this suggestion by FitzMaruice et al. (2017) is appropriate in this location of our manuscript.

9. Lines 354-355: It is worth mentioning that those exponents are related to the shape of the iceberg (in this case, taken as a flat plate).

The flat plate nature of tabular icebergs and ice islands is now mentioned.

10. Lines 386-387: "and the protection of a meltwater plume" - Again, it depends on the vertical velocity of the plume in relation to Δu (FitzMaurice et al., 2017). You could have a detached plume in a drifting iceberg.

We have added "and the **potential** protection of a meltwater plume" to account for the findings of FitzMaurice et al. (2017). We now also reference this work at this location.

**Technical corrections**

Line 16: "(...) thereby increasing the risk to marine transport and infrastructure as well as [affecting, impacting] the distribution of freshwater from the polar ice sheets."

This line now reads: ""(...) thereby increasing the risk to marine transport and infrastructure as well as affecting the distribution of freshwater from the polar ice sheets." We appreciate this catch and suggestion.

Lines 19-20: "The majority of thinning (73 %) resulted from basal ablation, but the associated volume loss was 12 times less than that caused by areal reduction" - It is not clear to me if the "associated volume loss" refers to the total thinning or only to the thinning caused by basal ablation.

We have clarified the wording so that it is apparent that the "associated volume loss" refers to the thinning caused by basal ablation.

Lines 34-35: "(...) and impact the biological and physical characteristics of ocean waters in their vicinity **due to meltwater input and latent heat uptake** resulting from their deterioration"

We have incorporated the reviewer's suggested edit.

Lines 57-61: "PII-A-1-f was a fragment of the 130 km$^2$ PII that calved from the Petermann Glacier in northwest Greenland on 5 August 2012 (Crawford et al., 2018a). After calving, the Canadian Ice Service (CIS; Environment and Climate Change Canada) tracked the ice island with RADARSAT-2 synthetic aperture radar (SAR) acquisitions. Between August 2012 and November 2014 the ice island drifted through Nares Strait and Baffin Bay, though it also experienced periods of stagnation while grounded in Kane Basin and northern Baffin Bay (Fig. 1a)" - I assume that the one that "calved from Petermann Glacier in northwest Greenland on 5 August 2012" is the PII. In this case, which ice island are you referring to in the next sentences? PII or its fragment PII-A-1-f?

We agree that it was not clear if we were referring to PII and PII-A-1-f. This has been clarified by adding some new information and slightly restructuring parts of the paragraph. It now reads, "PII-A-1-f was a fragment of the 130 km$^2$ PII that calved from Petermann Glacier in northwest Greenland on 5 August

2012 (Crawford et al., 2018a). Using tracking data derived from RADARSAT-2 satellite images in the Canadian Ice Island Drift Deterioration and Detection (CI2D3) Database (Crawford et al., 2018a) and by the Canadian Ice Service (CIS; Environment and Climate Change Canada), we were able to trace the origins of PII-A-1-f as PII broke up and drifted through Nares Strait and Baffin Bay between August 2012 and November 2014.  As this piece drifted south, it further fragmented and experienced periods of stagnation while grounded in Kane Basin and northern Baffin Bay (Fig. 1a). The PII-A-1-f fragment entered northern Baffin Bay in late 2013. Continued monitoring with RADARSAT-2 acquisitions showed that a portion of the deterioration that PII-A-1-f experienced after 2013 was caused by sidewall notches that progressively enlarged on opposing sides of the ice island."

Line 137: "ablation rate ($M_b$; m d$^{-1}$)" - there is a minus sign missing from d exponent

Fixed

Line 139: "$C$ (m$^{2/5}$ s$^{1/5}$ °C$^{-1}$)" - there is a minus sign missing from °C exponent

Fixed

Line 142: "melting point ($M_p$; °C)" - remove semicolon after °C

Removed

Equation 1: using fractional exponents makes the layout of this equation confusing (it looks like ΔT is part of the 4/5). I suggest using the typical notation: $M_b = 86400 C \Delta u^{0.8} \frac{\Delta T}{L^{0.2}}$.

We agree and have modified the layout of the equation.

Line 150: I would rephrase that as "Values of $C$ were obtained for each calibration interval i ($C_i$)", since up to that point you have only defined $C$.

We agree and have made this change.

Line 180: I think you meant "it is the time between the recording of the air and reflect**ed** waves by the receiver"

Fixed

Line 184: "However, an insufficient number (...)" - Since you use "However," again on line 187, you could remove this one without losing any meaning.

The first "However" on line 184 was removed.

Line 195: "the locations recorded by the mIPR onboard GPS were replaced with those recorded by a Hiper V dual-frequency GPS" - Remove first "GPS"

"GPS" was included in this sentence three times. We have removed the first "GPS" but have kept the latter two as they are referring to specific sensors.

Lines 267-268: "were an order-of-magnitude larger than those that have been previously calibrated or theoretically derived (Weeks and Campbell, 1986; White et al., 1980)." - I think you should mention here what the theoretically derived value is, so the reader can readily compare it with the following $C_i$ values.

The theoretically derived values are now reported, and the mention of 'previously calibrated' has been removed. We also corrected the publication year of the Weeks and Campbell work to 1973.

Line 308: "The magnitude of thinning (3 to 4 m) **observed** over this thin section **observed** along transect segment AB" - Perhaps replace the second "observed" with "present"

Replacement made

Lines 344 and 347: "(...) model skill would likely improve if Eq. (2) was calibrated for drifting vs. grounded ice islands. (...) different parameterizations required for predicting the basal ablation of drifting versus grounded ice islands" – I would change the first sentence (even remove it) to avoid repeating information a few lines below.

The portion of the first sentence that the reviewer notes has been removed to avoid repetition.

Line 375: improv**ed**

Fixed

Line 388: "(...) also contributed to the high, 13.5 m month$^{-1}$ basal ablation rate that **Jansen et al. (2007)** estimated (...)"

This citation has been added.

Line 409: "that on-ice data." - ... were collected (?)

Correct, this is now fixed and reads, "that on-ice data were collected".

Table 1: Check the "Dates" column on the first and second ablation periods. I believe it should be 2015-12-04 and 2015-12-05 instead of 2016.

Correct! We thank the reviewer for catching this and have fixed the dates.

Table 3 caption: "over the time period that basal ablation was derived with each incremental increase in the value assigned to a given variable." - I found this sentence very confusing

We have edited this caption. The concerned sentence now reads, "The sensitivity of the model was assessed as the mean percent increase in predicted cumulative basal ablation following each incremental increase in the value assigned to a given variable. The sensitivity was analysed over the longer period (2015-11-15 to 2016-09-18) due to the certainty in the bulk basal ablation magnitude over this time."

Figure 5 caption, line 642: There is a "(" missing from "b)"

Fixed

**Author changes:**

Greg Crocker's affiliation was corrected to Carleton University.

In the Abstract, "The calibration of the basal ablation model, the with such field data*….*" has been changed to "The calibration of the basal ablation model, the first known to be conducted with field data…"

We changed "deployment" to "collection of a long-term *in situ* dataset of ice island thinning" in the last paragraph of the Introduction.

The reference to Mingo et al. (forthcoming) has been updated in section 3.1 and References.

Minor edits to language and sentence structure have been made in section 3.2.1. We have added a reference to Oziel et al. (2019) in regard to our ocean velocity measurements in the same section. We also explicitly state that keel depth was derived using the measured ice thickness.

Edits have been made to ensure that 'data' are plural

$T_A$ was changed to $T_a$ in the Results to maintain consistency throughout the paper.

We explicitly state the length of time that basal ablation was estimated in section 4.1. A redundant sentence at the end of this section was removed as well.

We have modified the language at the end of section 4.2.1 to reflect our decision (not recommendation) to apply corrections to the oceanographic data.

Weeks et al. (1973) was corrected to Weeks and Campbell (1973) at the start of section 5.1.

An additional Løset (1993) reference has been added in regards to the assignment of ice temperature values in section 5.1. The two Løset (1993) reference are now assigned 'a' and 'b'. We now cite and reference FitzMaurice et al. (2016) and FitzMaurice and Stern (2018) in the same section.

We have corrected a reference to Ballicater (2012) to Ballicater Consulting (2012) at the end of section 5.2.

The reference to Sazidy et al. (forthcoming) has been updated in section 5.2 and References.

"…after an ice island stops freely drifting…" was modified to "…after an ice island stops drifting freely…" in the last paragraph of section 5.2.

A small amount of text was removed from the Conclusions as it was redundant with the preceding section.

---

## Author Comment (AC3) · 15 Dec 2019

**Reply to Reviewer 2 and other author changes: Manuscript #2019-125 by Crawford, Mueller, Crocker, Mingo, Desjardins, Dumont and Babin:** "*Ice island thinning: Rates and model calibration with in situ observations from Baffin Bay, Nunavut*"

Original reviewer comments are in black.

Author replies are in blue.

This paper presents measured ablation rates of a tabular iceberg which calved from the Peterman Glacier, North Greenland, which are then used to calibrate a model for melt rates. The *in situ* data set, which comprises repeated ice-penetrating radar surveys, surface mass balance measurements, as well as oceanographic data nearby is very unique, and a valuable contribution to the cryosphere community. Previously, studies on iceberg ablation rates were mostly based on modelling alone, or estimation of ablation rates from satellite altimetry, which is not straight forward because of the unknown surface processes and densification rates of a possible firn layer, which is especially important for the Antarctic icebergs. Thus, I would see it as an excellent contribution to The Cryosphere.

We thank the reviewer for their comments and their appreciation for the unique data set and basal ablation analysis presented in our manuscript.

I would like to rise a few points which might need addressing before publication:

1. At some point (section 3.2.2) it is stated that for the calibration of the model it is assumed that delta u is set equal to the ocean current velocity. As the iceberg is grounded during much of the time this might be a valid assumption. Nevertheless, as the iceberg can be easily tracked by remote sensing data, it would have been possible to look at real drift velocities, and how they compare to the ocean currents. Some studies have suggested (e.g. Lichey & Hellmer, 2001, Jounal of Glaciology) that not only the ocean current is responsible for iceberg drift velocity and direction, but also the wind conditions. A comparison / discussion of these parameters would be an improvement.

We agree with the reviewer, it is the relative speed between the ocean and drifting ice island that is necessary to constrain, and this will be affected by wind speed and direction. It is indeed possible to track ice islands with remote sensing data, and members of the Water and Ice Research Lab at Carleton University made a tremendous effort to do so while creating the Canadian Ice Island Drift, Deterioration and Detection (CI2D3) Database. This database will be used in the future to take on investigations into ice island drift patterns, such as that suggested by the reviewer. We believe that it is much too big of a step to incorporate such an assessment into the current paper. Such a study would also reply upon coarse resolution data for ice island drift velocities and modelled environmental conditions, which deviates from the *in situ* measurements and ocean conditions used in our assessment. We have added a statement to section 5.1 that notes the influence of both wind and ocean currents on iceberg drift and $\Delta u$ and reference the work by Lichey and Hellmer (2001) brought forward by the reviewer as well as Kubat et al. (2005).

2. In the introduction it is stated that this study is the first of its kind, for Arctic and Antarctic icebergs. However, to my knowledge there was a similar study set up for an Antarctic iceberg (Scambos at al., 2008, Journal of Glaciology) which might deserve a mentioning here.

Scambos et al. (2008) set up automated meteorological stations on two Antarctic icebergs, and paired one station with a stationary ice penetrating radar. Unfortunately, their ice thickness data was unusable as a result of interference from nearby instruments. Due to this, our dataset still includes the first long-term *in situ* ice thickness measurements from an iceberg in either Polar Region. We also present the first spatially distributed thickness change values, which were possible with our repeated mobile ice penetrating radar

transects. The study by Scambos et al. (2008) incorporated novel field data and we agree that the work deserves mentioning as it is an important pre-cursor to our work. We have included text that describes this work in our Discussion (section 5.2).

3. In the discussion the big difference between the ration of basal and surface ablation rates from results of a former study is mentioned. When comparing these results it has to be considered the in case of the other study the Antarctic tabular iceberg started off with a firn column, while the Peterman iceberg did not have any snow cover, so in fact a blue-ice surface. If there is a firn column, surface melt water can percolate into the firn and refreeze, so the mass is not immediately lost. While on blue ice it is more likely to run off. The problem of refreezing melt water and firn densification is the biggest contribution to uncertainty for previous studies estimating ablation of tabular icebergs from altimeter data. For this setting it would be immensely helpful to have an in situ data set like the one presented here. This might be added to the discussion.

We agree that it is important to note the difference in surface conditions between our study subject and the large tabular icebergs in Antarctica. This is included in section 5.2. With this note, we also mention the iceberg firn observations of Scambos et al. (2008) and suggest that a follow-up study with concurrent ice thickness measurements would be of high value.

**Author changes:**

Greg Crocker's affiliation was corrected to Carleton University.

In the Abstract, "The calibration of the basal ablation model, the with such field data...." has been changed to "The calibration of the basal ablation model, the first known to be conducted with field data…"

We changed "deployment" to "collection of a long-term *in situ* dataset of ice island thinning" in the last paragraph of the Introduction.

The reference to Mingo et al. (forthcoming) has been updated in section 3.1 and References.

Minor edits to language and sentence structure have been made in section 3.2.1. We have added a reference to Oziel et al. (2019) in regard to our ocean velocity measurements in the same section. We also explicitly state that keel depth was derived using the measured ice thickness.

Edits have been made to ensure that 'data' are plural

$T_A$ was changed to $T_a$ in the Results to maintain consistency throughout the paper.

We explicitly state the length of time that basal ablation was estimated in section 4.1. A redundant sentence at the end of this section was removed as well.

We have modified the language at the end of section 4.2.1 to reflect our decision (not recommendation) to apply corrections to the oceanographic data.

Weeks et al. (1973) was corrected to Weeks and Campbell (1973) at the start of section 5.1.

An additional Løset (1993) reference has been added in regards to the assignment of ice temperature values in section 5.1. The two Løset (1993) reference are now assigned 'a' and 'b'. We now cite and reference FitzMaurice et al. (2016) and FitzMaurice and Stern (2018) in the same section.

We have corrected a reference to Ballicater (2012) to Ballicater Consulting (2012) at the end of section 5.2.

The reference to Sazidy et al. (forthcoming) has been updated in section 5.2 and References.

"…after an ice island stops freely drifting…" was modified to "…after an ice island stops drifting freely…" in the last paragraph of section 5.2.

A small amount of text was removed from the Conclusions as it was redundant with the preceding section.